# Phagosomal removal of fungal melanin reprograms macrophage metabolism to promote antifungal immunity

Samuel M. Gonçalves[1,2], Cláudio Duarte-Oliveira[1,2], Cláudia F. Campos[1,2], Vishukumar Aimanianda[3], Rob ter Horst [4], Luis Leite[5], Toine Mercier [6,7], Paulo Pereira [8], Miguel Fernández-García[9], Daniela Antunes [1,2], Cláudia S. Rodrigues[1,2], Catarina Barbosa-Matos [1,2], Joana Gaifem[1,2], Inês Mesquita[1,2], António Marques[10], Nuno S. Osório [1,2], Egídio Torrado [1,2], Fernando Rodrigues[1,2], Sandra Costa[1,2], Leo AB. Joosten [4], Katrien Lagrou [7,11], Johan Maertens[6,7], João F. Lacerda[8], António Campos Jr.[5], Gordon D. Brown[12], Axel A. Brakhage[13,14], Coral Barbas [9], Ricardo Silvestre [1,2], Frank L. van de Veerdonk[4], Georgios Chamilos[15,16], Mihai G. Netea [4,17], Jean-Paul Latgé [3], Cristina Cunha [1,2,18] & Agostinho Carvalho [1,2,18✉]

In response to infection, macrophages adapt their metabolism rapidly to enhance glycolysis and fuel specialized antimicrobial effector functions. Here we show that fungal melanin is an essential molecule required for the metabolic rewiring of macrophages during infection with the fungal pathogen *Aspergillus fumigatus*. Using pharmacological and genetic tools, we reveal a molecular link between calcium sequestration by melanin inside the phagosome and induction of glycolysis required for efficient innate immune responses. By remodeling the intracellular calcium machinery and impairing signaling via calmodulin, melanin drives an immunometabolic signaling axis towards glycolysis with activation of hypoxia-inducible factor 1 subunit alpha (HIF-1α) and phagosomal recruitment of mammalian target of rapamycin (mTOR). These data demonstrate a pivotal mechanism in the immunometabolic regulation of macrophages during fungal infection and highlight the metabolic repurposing of immune cells as a potential therapeutic strategy.

[1] Life and Health Sciences Research Institute (ICVS), School of Medicine, University of Minho, 4710-057 Braga, Portugal. [2] ICVS/3B's - PT Government Associate Laboratory, Guimarães/Braga, Portugal. [3] Unité des Aspergillus, Institut Pasteur, 75015 Paris, France. [4] Department of Internal Medicine and Radboud Center for Infectious Diseases, Radboud University Medical Centre, 6500HB Nijmegen, Netherlands. [5] STMO, Instituto Português de Oncologia, 4200-072 Porto, Portugal. [6] Department of Hematology, UZ Leuven, 3000 Leuven, Belgium. [7] Department of Microbiology and Immunology, KU Leuven, 3000 Leuven, Belgium. [8] Instituto de Medicina Molecular, Faculdade de Medicina de Lisboa, 1649-028 Lisboa, Portugal. [9] Center for Metabolomics and Bioanalysis, Faculty of Pharmacy, San Pablo CEU University, 28668 Madrid, Spain. [10] Serviço de Imuno-Hemoterapia, Hospital de Braga, 4710-243 Braga, Portugal. [11] Department of Laboratory Medicine, UZ Leuven, 3000 Leuven, Belgium. [12] MRC Centre for Medical Mycology, University of Aberdeen, Aberdeen AB25 2ZD, UK. [13] Department of Molecular and Applied Microbiology, Leibniz-Institute for Natural Product Research and Infection Biology, 07745 Jena, Germany. [14] Institute of Microbiology, Friedrich Schiller University, 07743 Jena, Germany. [15] School of Medicine, University of Crete, 70013 Heraklion, Greece. [16] Institute of Molecular Biology and Biotechnology, FORTH, 70013 Heraklion, Greece. [17] Department of Genomics & Immunoregulation, Life and Medical Sciences Institute, University of Bonn, 53115 Bonn, Germany. [18]These authors contributed equally: Cristina Cunha, Agostinho Carvalho. ✉email: agostinhocarvalho@med.uminho.pt

The reprogramming of cellular metabolism is a fundamental mechanism through which innate immune cells meet the energetic and anabolic needs during host defense against invading pathogens[1]. Sensing of microbial ligands drives the upregulation of glycolysis, which delivers a rapid source of energy to support antimicrobial functions and the production of cytokines[2]. The enhanced glycolytic activity also directly supports cytokine expression through mechanisms that involve moonlighting activities of the enzymes themselves[3,4]. The metabolic pattern of myeloid cells activated by canonical stimuli (e.g., lipopolysaccharide, LPS) generally implies the downregulation of mitochondrial carbon metabolism and oxidative phosphorylation[5–8]. The disruption in the tricarboxylic acid cycle promotes the accumulation of mitochondrial metabolites that in turn regulate the expression of glycolytic enzymes to support inflammatory responses and antimicrobial functions[9]. This metabolic reprogramming generates lactate from glucose without further mitochondrial oxidation, despite normoxic conditions, a phenotype observed in cancer cells and known as the Warburg effect[10]. It is now clear that, under conditions of microbial challenge, glucose metabolism is critically required; for example, compounds that block the metabolic shift to glycolysis (e.g., inhibitors of the mammalian target of rapamycin (mTOR) pathway or metformin) restrain cytokine production and dampen immune responses, hampering pathogen clearance[9,11–13]. Importantly however, other studies have shown that stimuli other than LPS induce upregulation of both glycolysis and oxidative phosphorylation in immune cells[14].

The recognition of pathogen-associated molecular patterns (PAMPs) drives substantial changes in cellular metabolism and effector functions of immune cells[10]. Owing to its dynamic composition and structural plasticity, the cell wall is considered the most relevant repository for fungal PAMPs[15,16]. Exposure to β-1,3-glucan has been shown to promote the metabolic reprogramming of monocytes leading to a trained immunity phenotype characterized by enhanced cytokine production in response to heterologous secondary stimulation[11,17–19]. The requirement for the metabolic rewiring of myeloid cells was also demonstrated in vivo, since the pharmacological impairment of glycolysis[20] or the blockade of metabolic pathways with metformin[21] increased susceptibility of mice to systemic candidiasis. In turn, fungal pathogens have evolved intricate virulence strategies to withstand the host immune response, by exploiting nutritional weaknesses of immune cells leading to their death[22].

Although β-1,3-glucan is a major fungal cell wall component, not much is known about the metabolic regulation of immunity to fungi other than C. albicans and whether other cell wall polysaccharides participate in these signaling events. Here, we investigated the immunometabolic response of macrophages to the opportunistic fungal pathogen Aspergillus fumigatus. This fungus can cause a wide spectrum of diseases with distinct clinical manifestations[23]. Epidemiological data have revealed that A. fumigatus causes >200,000 invasive infections each year in hematological patients under aggressive chemotherapy or undergoing solid organ or allogeneic stem-cell transplantation[24]. Because there are no licensed vaccines and the currently available diagnostic tests lack accuracy, mortality rates after infection are estimated above 30%[25].

Macrophages are considered critical in preventing fungal germination and tissue invasion early after infection, particularly before the influx of neutrophils[26]. One relevant mechanism is represented by the instruction of programmed necrosis in macrophages by calcineurin, which by stimulating the lateral transfer of conidia between macrophages, enables the control of fungal germination[27]. Other studies have highlighted the importance of inflammatory monocytes in experimental aspergillosis through their ability to orchestrate the conidiacidal activity of neutrophils

and dendritic cells[28]. Moreover, there are several examples of genetic variants that predispose humans to aspergillosis by affecting the ability of myeloid cells to produce cytokines or exert their killing activity[29–32]. Although the fine-tuned regulation of cellular metabolism is required for the functional activity of macrophages, how these processes are orchestrated in response to A. fumigatus remains undefined.

Here, we sought to understand the mechanisms through which infection with A. fumigatus rewires macrophage metabolism toward efficient innate immune responses. We show that fungal melanin is an essential PAMP required for the Warburg shift and the ensuing immunometabolic responses in macrophages. Our results define how the host is able to counter the immune inhibitory mechanisms deployed by fungal melanin in order to promote efficient antifungal immune responses required to control infection.

## Results

**Macrophage metabolism is modulated by *A. fumigatus* infection.** Our first aim was to investigate host cellular metabolism during the interaction of human macrophages with A. fumigatus. We chose two time points after infection corresponding to the early transcriptional events following phagocytosis (2 h) and the initial phases of fungal escape by germination (6 h). RNA-seq analysis revealed substantial changes in the macrophage transcriptome after 2 h of infection, encompassing 1187 differentially expressed genes relative to uninfected controls (Fig. 1a). The number of differentially expressed genes decreased markedly at 6 h to a total of 83, indicating that the bulk of transcriptional changes occurred early after infection. Enrichment analysis demonstrated that pathways involved in cytokine production and signaling, response to inflammatory stimuli, and immune cell differentiation were overrepresented in the upregulated genes after 2 h of infection (Fig. 1b). In accordance with the low number of differentially expressed genes detected 6 h after infection, genes involved in the negative regulation of transcription and DNA binding were enriched in the downregulated pathways.

A targeted analysis of the differentially expressed genes after 2 h of infection using hallmark gene sets from the Molecular Signatures Database[33] revealed a substantial upregulation of genes involved in glycolysis, but not oxidative phosphorylation (Fig. 1c). The commitment of macrophages toward glycolysis was reflected by the upregulation of the glucose transporters SLC2A1 (GLUT1) and SLC2A6 (GLUT6), the key glycolytic regulator 6-phosphofructo-2-kinase/fructose-2,6-bisphosphatase 3 (PFKFB3), and hypoxia-inducible factor 1 subunit alpha (HIF1A). In contrast, thioredoxin-interacting protein (TXNIP), which encodes a redox sensor that suppresses glucose uptake and metabolism[34], was instead downregulated. These results were consistent with a skewing of infected macrophages toward an M1 inflammatory phenotype characterized by the upregulation of several proinflammatory cytokines, including IL1B, TNF, and IL-6.

Gene expression analysis using a more detailed time course of infection demonstrated that induction of glycolytic genes such as GLUT1, hexokinase 2 (HK2), and PFKFB3 initiated early after the challenge, and was sustained throughout infection (Fig. 1d). Glut1 and several glycolytic enzymes were also induced in the lungs of mice as early as day 1 and until day 3 after fungal infection (Fig. 1e). The transcriptional induction of glycolysis was confirmed by the increased secretion of lactate and glucose consumption by human macrophages throughout the infection, an effect that was dependent on the multiplicity of infection (Fig. 1f), but was not influenced by the differentiating stimulus (Supplementary Fig. 1a). Fungal infection also elicited an increased ADP/ATP balance (Fig. 1g). To gain further insight into the metabolites produced

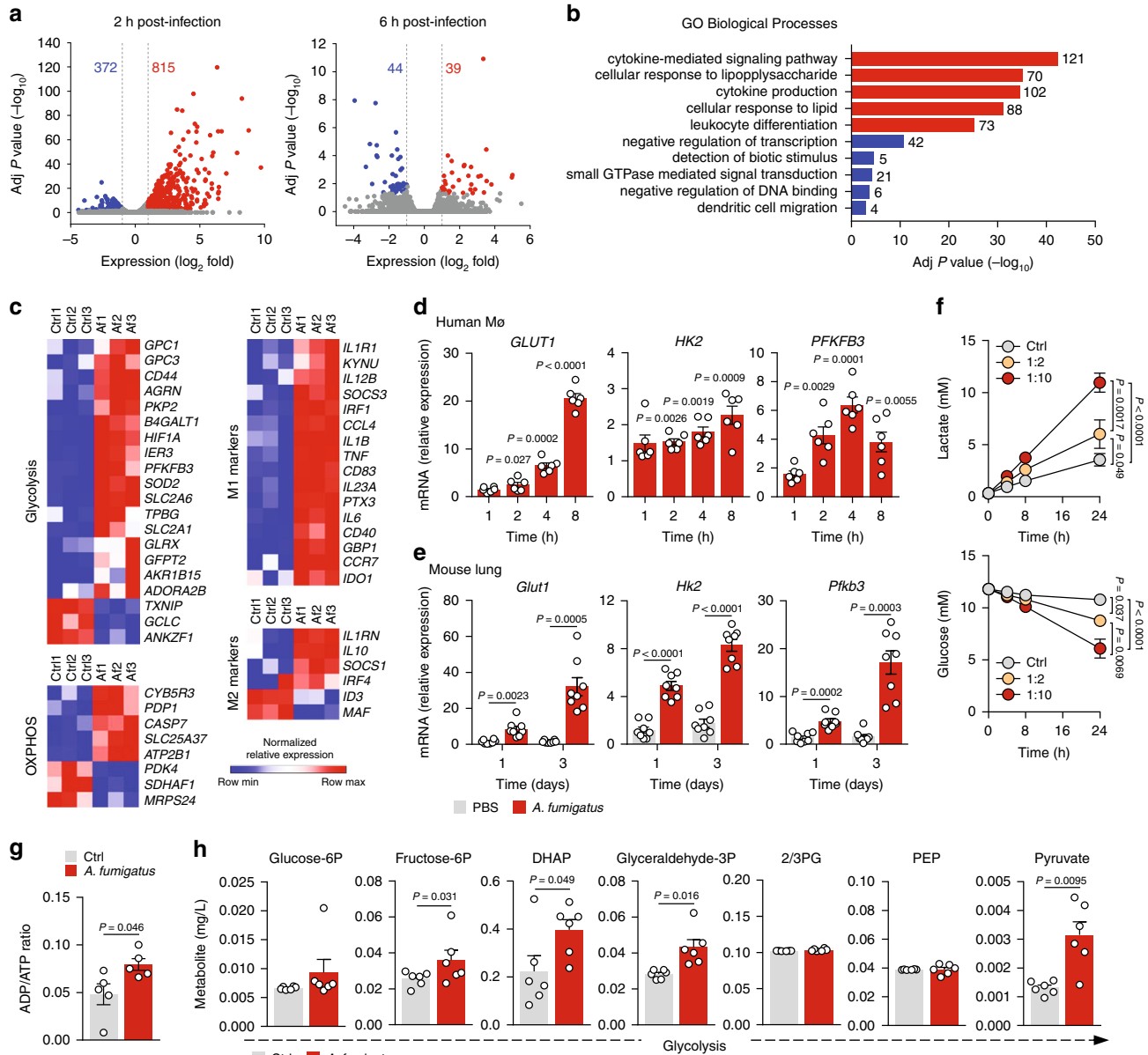

**Fig. 1 A. fumigatus induces glycolysis in macrophages. a** Transcriptome analysis of human macrophages infected with *A. fumigatus* for 2 or 6 h. Numbers indicate genes with differential expression, up- (red) or downregulated (blue) in infected relative to uninfected cells. **b** Pathway analysis of up- (red) or downregulated (blue) genes 2 h after infection. Genes were categorized into the most represented pathways, in which the gene products are involved. **c** Transcriptional profiles of macrophages left untreated (Ctrl) or infected with *A. fumigatus* (Af) for 2 h (*n* = 3). Expression of genes is presented as centered, and scaled log₂ fluorescence intensity (blue and red keys) grouped by product function. **d** mRNA expression of *GLUT1*, *HK2*, and *PFKFB3* in macrophages infected for 1, 2, 4, or 8 h relative to uninfected cells (*n* = 6). **e** mRNA expression of *Glut1*, *Hk2*, and *Pfkfb3* in mouse lungs sampled 1 or 3 days after infection (*n* = 8, representative of three independent experiments). **f** Lactate secretion and glucose consumption by macrophages left untreated or infected at 1:2 or 1:10 for 24 h (*n* = 6). **g** ADP/ATP ratio (*n* = 5) and **h** targeted metabolomics (*n* = 6) in macrophages left untreated or infected for 6 h. Data are expressed as mean values ± SEM. *P*-values were calculated using Student's two-tailed *t* test or two-way ANOVA with Tukey's multiple comparisons test.

along the glycolytic pathway, we performed a targeted analysis of metabolic pathways after 6 h of infection using liquid chromatography tandem-mass spectrometry (LC–MS/MS). In these conditions, increased levels of most glycolytic intermediates were detected, namely, fructose-6-phosphate, dihydroxyacetone phosphate, glyceraldehyde-3-phosphate, and pyruvate (Fig. 1h).

The metabolic rewiring of macrophages during infection commits them to glucose metabolism for survival, and this phenotype is exploited by *C. albicans* to trigger cell death by depleting glucose levels[22]. In contrast, human macrophages

infected with *A. fumigatus* did not display a significant loss of viability, even after 24 h of infection (Supplementary Fig. 1b). Likewise, the total number of macrophages in the lungs of infected mice also remained intact after infection (Supplementary Fig. 1c). Further supporting the lack of macrophage death due to competition for glucose, expression of fungal glycolytic enzymes, including ATP-dependent 6-phosphofructokinase (*pfkA*) and hexokinase-1 (*hxkA*), was not significantly modulated during infection (Supplementary Fig. 1d), a finding in line with the negligible amount of lactate measured during fungal culture

(Supplementary Fig. 1e). These results reveal the glycolytic reprogramming of macrophages in response to *A. fumigatus*, while highlighting different metabolic strategies across fungal genera to survive and thrive during infection.

**Glycolysis is required for immune responses to *A. fumigatus*.** Production of proinflammatory cytokines, phagocytosis and killing are major effector functions of macrophages during infection with *A. fumigatus*[16]. We tested therefore how the inhibition of glycolysis using 2-deoxyglucose (2-DG), a competitive inhibitor of hexokinase, affected these processes. We confirmed that the treatment with 2-DG decreased lactate secretion by infected macrophages in a dose-dependent manner (Fig. 2a), without affecting cellular viability (Supplementary Fig. 2a). Likewise, macrophages cultured in glucose-deprived media or media containing galactose, impairing the glycolytic flux, were also unable to secrete lactate during infection (Supplementary Fig. 2b). Upon treatment with 2-DG (Fig. 2b) or using media without glucose or with galactose (Supplementary Fig. 2c), macrophages displayed an impaired conidiacidal activity, which was in line with the decreased production of reactive oxygen species (Fig. 2c). The phagocytic ability instead remained intact (Fig. 2d), a finding supported by data showing that phagocytosis relies on oxidative phosphorylation[14]. In addition, treatment of macrophages with 2-DG (Fig. 2e) or the use of glucose-deprived media (Supplementary Fig. 2d) impaired the production of proinflammatory cytokines, including IL-1β, TNF, and IL-6, after 24 h of infection. The production of adaptive cytokines by PBMCs, such as IFNγ and IL-17A, detected after 7 days of infection, was also compromised following the blockade of glycolysis (Supplementary Fig. 2e).

Since the blockade of hexokinase by 2-DG might also affect the pentose phosphate pathway (PPP), we evaluated lactate secretion by macrophages treated with 3-(3-pyridinyl)-1-(4-pyridinyl)-2-propen-1-one (3PO), a selective inhibitor of PFKFB3, and 6-aminonicotinamide (6-AN), an inhibitor of the 6-phosphogluconate dehydrogenase enzyme from the PPP. In these conditions, we confirmed an impaired lactate secretion (Fig. 2f) and cytokine production (Supplementary Fig. 2f) using 3PO, but not 6-AN. To gain further insight into lung microenvironment-associated effects on glycolysis, we measured lactate secretion by murine alveolar macrophages after infection (Fig. 2g; Supplementary Fig. 2g). The results showed an induction of lactate secretion, an effect that was abrogated following treatment with 2-DG. Collectively, these data highlight the critical requirement for glycolysis to host defense mechanisms in response to *A. fumigatus*.

Because our results pointed to a pivotal role of glucose metabolism in antifungal host defense, we next validated this requirement in a mouse model of pulmonary aspergillosis, in which immunocompetent animals were treated with 2-DG or PBS prior to infection (Fig. 2h). In line with the activation of glucose metabolism, we detected systemic hypoglycemia in mice after the challenge (Supplementary Fig. 2h). Blocking glycolysis rendered mice more susceptible to infection, as revealed by the increased fungal burden in the lungs 1 day after infection (Fig. 2h), a finding that did not involve loss of viability of alveolar macrophages (Supplementary Fig. 2i). In addition, the levels of cytokines in lung homogenates from 2-DG-treated mice were significantly lower than those from mock-treated animals (Fig. 2i). Consistently, the production of cytokines after restimulation of splenocytes from 2-DG-treated mice (Fig. 2j) and their conidiacidal activity (Fig. 2k) was also impaired. Taken together, these data confirm that glucose metabolism plays a central role in the induction of host responses to *A. fumigatus* in vivo.

**Induction of host glycolysis depends on fungal melanin.** To identify the mechanism(s) underlying the activation of glycolysis during infection, we compared lactate secretion by macrophages challenged with different fungal morphotypes for 24 h. Secretion of lactate was higher when stimulation involved dormant conidia compared with swollen conidia or germ tubes (Fig. 3a). We hypothesized that the fungal molecule(s) inducing glycolysis were present on dormant conidia, but were mostly lost after germination. Rodlet and melanin, made up of hydrophobic protein RodA and polymerized 1,8-dihydroxynaphthalene (DHN), respectively, form the outermost layer of dormant conidia and are both removed during germination[35]. To test their contribution, we assessed lactate secretion after infection with dormant conidia of the ΔrodA mutant devoid of the surface rodlet layer[36] or the ΔpksP mutant lacking the polyketide synthase responsible for the initial step in DHN–melanin biosynthesis[37]. Infection with ΔrodA conidia induced lactate secretion (Fig. 3b) and glucose consumption (Supplementary Fig. 3a) to an extent comparable with that measured for the parental strain Δku80, whereas ΔpksP or ΔrodA/pksP conidia failed to efficiently trigger this metabolic reprogramming. Macrophages infected with ΔpksP conidia also displayed decreased mRNA levels of glycolysis-related genes (Fig. 3c). Collectively, these findings are consistent with a critical role for fungal melanin in reprogramming macrophage metabolism during infection.

To determine at which stage the putative ligand(s) inducing glycolysis were synthesized in the pathway, we screened deletion mutants in the enzymes required to catalyze each step for their ability to activate glycolysis. We confirmed that lactate secretion was severely affected throughout the entire DHN–melanin biosynthetic pathway (Supplementary Fig. 3b), suggesting that fully mature melanin particles, and not the heptaketide naphthopyrone synthesized by PksP alone, is required for the metabolic reprogramming. The requisite role for melanin in the activation of glycolysis was confirmed by similar defective levels of lactate secreted by macrophages infected with the albino strains of *A. fumigatus* CBS 110.46 and CBS 386.75[38] (Fig. 3d). To elucidate whether melanin pigments that differ from DHN–melanin could also activate glycolysis, we infected macrophages with *A. nidulans* whose melanin differs from that of *A. fumigatus*[39]. In these conditions, lactate secretion occurred to a degree similar to that induced by the Δku80 strain of *A. fumigatus* (Fig. 3e).

Because the recently identified melanin receptor MelLec could be involved in melanin-induced activation of glycolysis, we stimulated macrophages with 1,8-DHN, polymerization of which leads to melanin biosynthesis, and that contains the conserved naphthalene-diol unit recognized by MelLec[32]. We observed that the levels of lactate secreted by macrophages stimulated with 1,8-DHN were significantly lower than those after infection with the Δku80 strain (Fig. 3f), suggesting that recognition of 1,8-DHN melanin and signaling via MelLec is, to a large extent, dispensable for the activation of glycolysis. In support of this, infection of bone marrow-derived macrophages (BMDMs), which lack expression of MelLec[32], with ΔpksP conidia also induced lower concentrations of secreted lactate (Supplementary Fig. 3c). Moreover, human macrophages carrying the loss-of-function rs2306894 variant in MelLec displayed comparable levels of lactate secreted after infection to that of wild-type cells (Supplementary Fig. 3d).

To directly link the presence of melanin on the conidial surface and activation of glycolysis, we stimulated macrophages with melanin ghosts alone or added together with ΔpksP conidia. Strikingly, none of these conditions significantly induced lactate secretion (Fig. 3g), a finding suggesting the need for melanin to be actively shed or removed during conidial germination to activate glycolysis. Indeed, only live, but not heat-killed or UV-

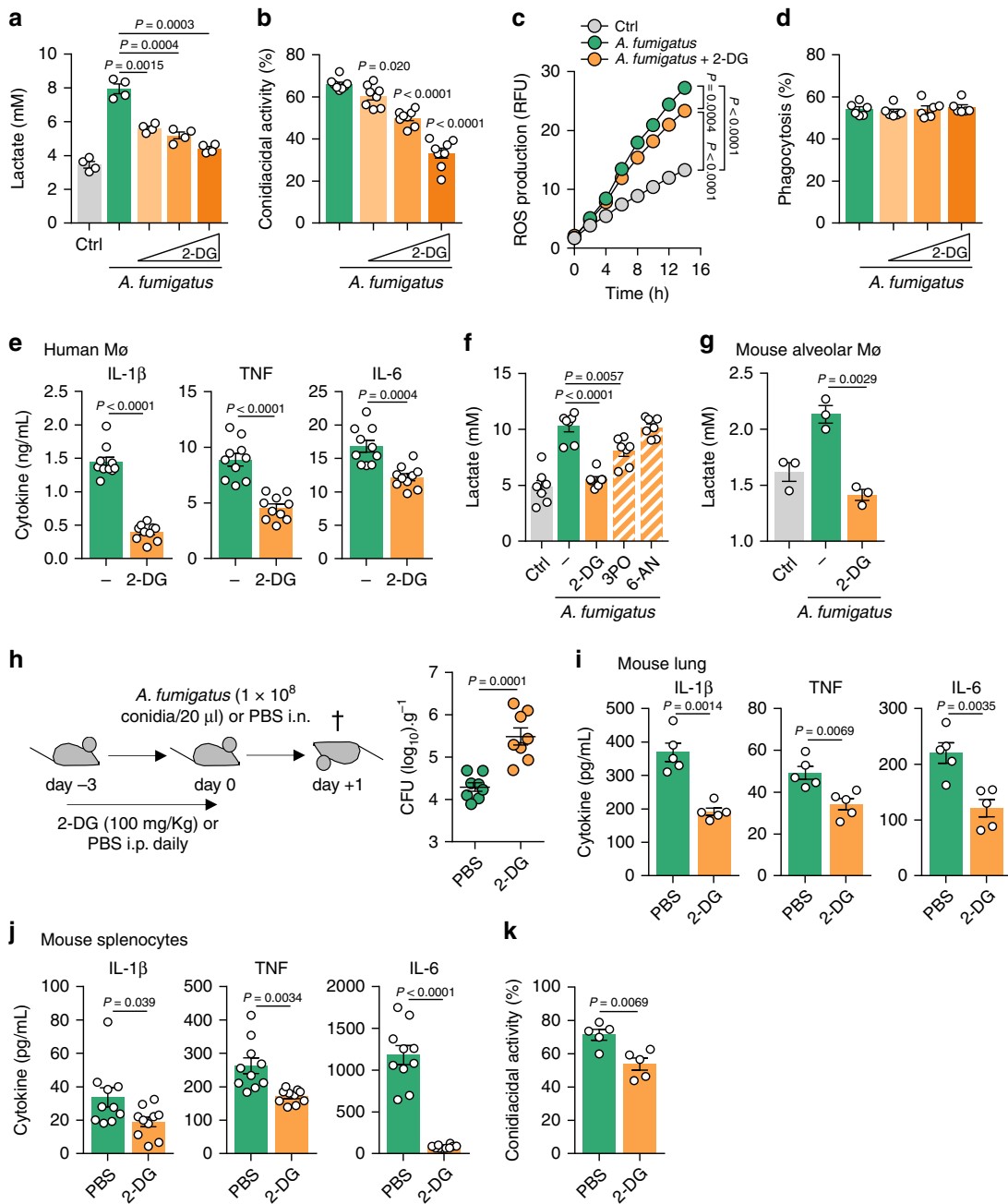

**Fig. 2 Glycolysis is required for antifungal immune responses. a** Lactate secretion ($n = 4$) and **b** conidiacidal activity ($n = 8$) of macrophages left untreated (Ctrl) or infected with *A. fumigatus* for 24 or 3 h, respectively, without or with 5, 10, or 20 mM 2-DG ($n = 4$). **c** ROS production by macrophages left untreated or infected for 24 h without or with 10 mM 2-DG ($n = 8$). **d** Phagocytosis of macrophages infected for 1 h without or with 5, 10, or 20 mM 2-DG ($n = 6$). **e** Production of IL-1β, TNF, and IL-6 by macrophages infected for 24 h without or with 10 mM 2-DG ($n = 10$). **f** Lactate secretion by macrophages left untreated (Ctrl) or infected with *A. fumigatus* for 24 h without or with 10 mM 2-DG, 30 μM 3PO or 500 nM 6-AN ($n = 7$). **g** Lactate secretion by mouse alveolar macrophages left untreated (Ctrl) or infected for 24 h without or with 10 mM 2-DG ($n = 3$). **h** Experimental setup for glycolysis inhibition in vivo during infection. Fungal burden (log10) per gram of lung tissue was determined in PBS- or 2-DG-treated mice after 1 day of infection ($n = 8$, representative of three independent experiments). **i** Levels of IL-1β, TNF, and IL-6 in lung homogenates of PBS- or 2-DG-treated mice after 1 day of infection ($n = 5$). **j** Production of IL-1β, TNF, and IL-6 and **k** conidiacidal activity of mouse splenocytes isolated from PBS- or 2-DG-treated mice and restimulated for 24 ($n = 10$) or 2 h ($n = 5$), respectively. Data are expressed as mean values ± SEM. *P*-values were calculated using Student's two-tailed *t* test or two-way ANOVA with Tukey's multiple comparisons test.

inactivated, conidia triggered lactate secretion by macrophages (Fig. 3h), a finding highly suggestive of a cell wall remodeling associated with melanin release. To further test this, we performed functional complementation experiments using live dormant Δ*rodA*/*pksP* conidia (allowing enhanced adhesion due to removal of the hydrophobic layer) coated with purified fragments

of melanin isolated from *A. fumigatus* conidia. The ability of the Δ*rodA*/*pksP* strain to induce lactate secretion was rescued by the melanin coating in a dose-dependent manner (Fig. 3i), a finding supported by the increase in the mRNA expression of glycolytic genes (Fig. 3j). In line with the link between glycolysis and cytokine production, infection with melanin-coated live

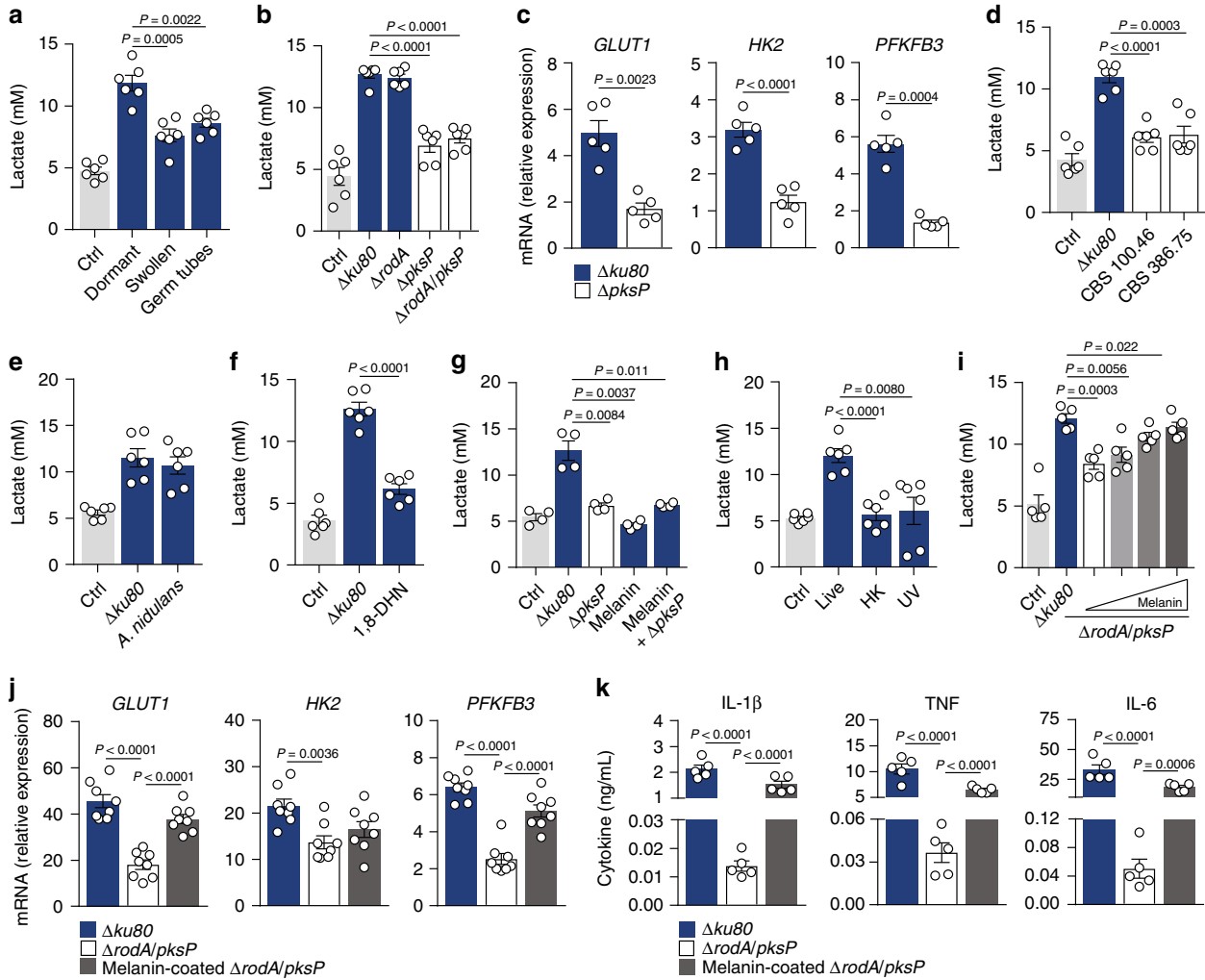

**Fig. 3 Intracellular removal of fungal melanin induces glucose metabolism. a** Lactate secretion by macrophages left untreated (Ctrl) or infected with dormant or swollen conidia, or germ tubes of *A. fumigatus* for 24 h (*n* = 6). **b** Lactate secretion by macrophages left untreated (Ctrl) or infected with dormant conidia from the parental Δ*ku80* strain or the cell wall mutants Δ*rodA*, Δ*pksP*, or Δ*rodA/pksP* for 24 h (*n* = 6). **c** mRNA expression of *GLUT1*, *HK2*, and *PFKFB3* in macrophages infected with Δ*ku80* or Δ*pksP* strains for 2 h relative to uninfected cells (*n* = 5). **d** Lactate secretion by macrophages left untreated (Ctrl) or infected with the albino CBS 100.46 and 386.75 strains of *A. fumigatus* (*n* = 6), **e** the WG355 strain of *A. nidulans* (*n* = 6), **f** 1,8-DHN (*n* = 6), or **g** the Δ*pksP* strain, DHN–melanin, or DHN–melanin added together with Δ*pksP* conidia (*n* = 4) for 24 h. Infection with Δ*ku80* conidia was used as control. **h** Lactate secretion by macrophages left untreated (Ctrl) or infected with live, heat-killed (HK), or UV-inactivated (UV) conidia for 24 h (*n* = 5). **i** Lactate secretion by macrophages infected with the Δ*ku80*, Δ*rodA/pksP*, or the Δ*rodA/pksP* strain coated with 100, 300, and 600 μg/mL of DHN–melanin for 24 h (*n* = 5). **j** mRNA expression of *GLUT1*, *HK2*, and *PFKFB3* (*n* = 8) and **k** production of IL-1β, TNF, and IL-6 (*n* = 5) by macrophages infected with the Δ*ku80*, Δ*rodA/pksP*, or the melanin-coated Δ*rodA/pksP* strain for 2 or 24 h, respectively, relative to uninfected cells (*n* = 8). Data are expressed as mean values ± SEM. *P*-values were calculated using Student's two-tailed *t* test or one-way ANOVA with Tukey's multiple comparisons test.

Δ*rodA/pksP* conidia also reverted the defective production of proinflammatory cytokines (Fig. 3k). The same experimental approach using inactivated conidia was instead unable to restore lactate secretion to normal levels (Supplementary Fig. 3e). These results demonstrate that the active intracellular removal of melanin is required for the reprogramming of glucose metabolism in macrophages.

**Fungal melanin rewires host metabolism via mTOR and HIF-1α.** The metabolic reprogramming of myeloid cells toward glycolysis is driven by the activation of mTOR, an intracellular sensor that functions as a master regulator of glucose metabolism[40]. Consistent with this, the transcriptome of human macrophages after 2 h of infection with *A. fumigatus* was markedly enriched in genes involved in mTOR signaling (Fig. 4a). The activation of the mTOR pathway during infection

was confirmed by the enhanced phosphorylation of the p70S6 kinase, a downstream target of the mTORC1 complex (Fig. 4b; Supplementary Fig. 4a). Remarkably, Δ*pksP* conidia failed to elicit a significant increase in p-p70S6K, indicating that melanin is required for the induction of mTOR signaling. Because mTOR activation is often mediated by the intermediary activation of the phosphatidylinositol-4,5-bisphosphate 3-kinase (PI3K)/Akt pathway[41], we assessed phosphorylation of Akt in the same conditions. Similar to mTOR, the levels of p-Akt were increased during infection with Δ*ku80* conidia, but less when the Δ*pksP* strain was used (Fig. 4b; Supplementary Fig. 4b). As these results suggest a coupled activation of Akt/mTOR and glycolysis, we next investigated the causality between these two processes during infection. Inhibiting this pathway with rapamycin (mTOR) (Fig. 4c) or wortmannin (Akt) (Supplementary Fig. 4c) during infection with Δ*ku80* conidia impaired the

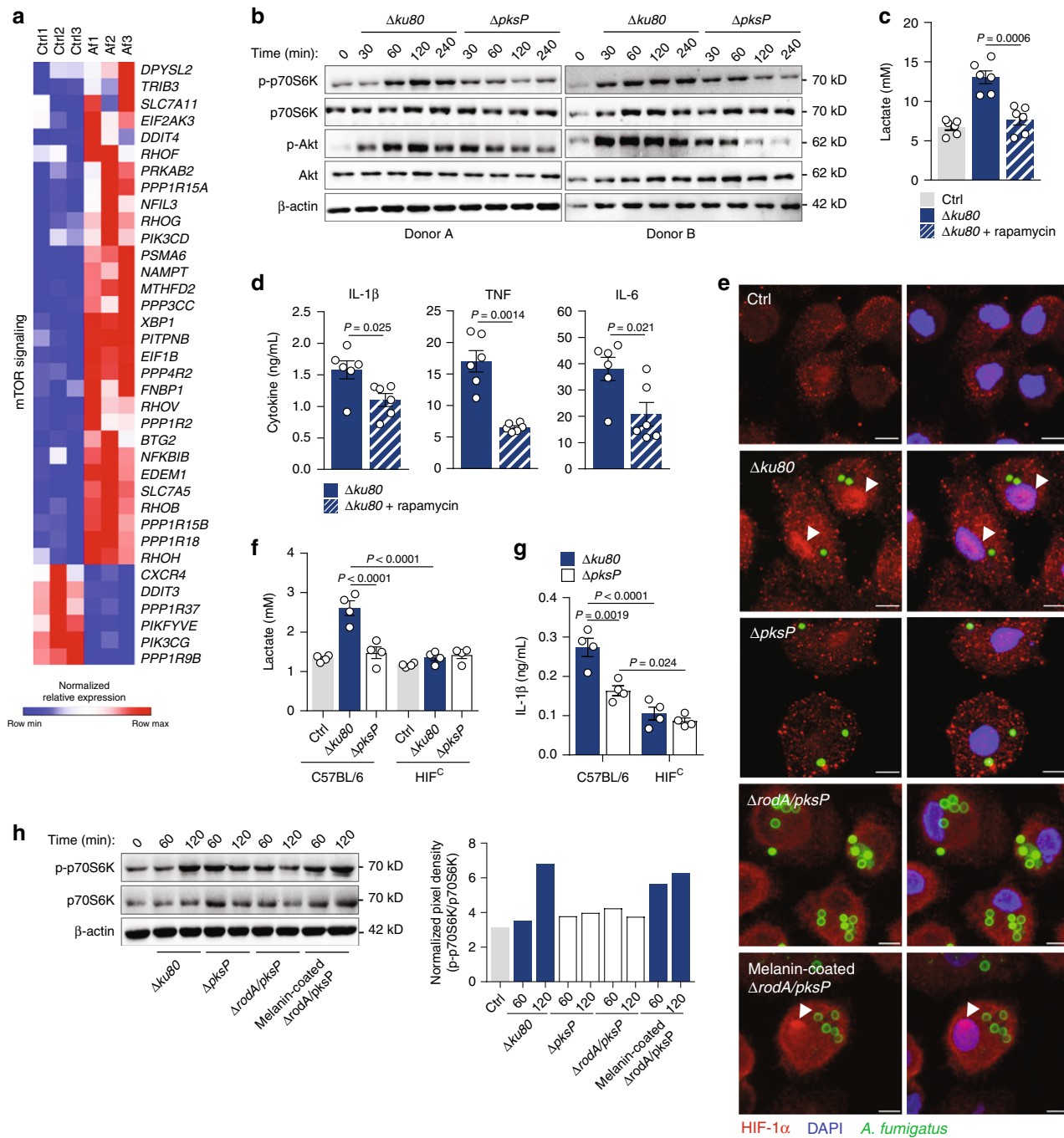

**Fig. 4 mTOR and HIF-1α reprogram metabolism in response to _A. fumigatus_. a** Transcriptional profiles of macrophages left untreated (Ctrl) or infected with _A. fumigatus_ (Af) for 2 h ($n = 3$). Expression of genes is presented as centered and scaled $\log_2$ fluorescence intensity (blue and red keys). **b** The total and p-p70S6K and total and p-Akt in macrophages infected with the Δ_ku80_ or Δ_pksP_ strains for 4 h (representative of three independent experiments, with β-actin used as loading control). **c** Lactate secretion and **d** production of IL-1β, TNF, and IL-6 by macrophages left untreated (Ctrl) or infected with the Δ_ku80_ strain for 24 h without or with 10 mM rapamycin ($n = 6$). **e** Expression of HIF-1α in macrophages left untreated (Ctrl) or infected with the Δ_ku80_, Δ_rodA/pksP_ or the melanin-coated Δ_rodA/pksP_ strain for 2 h (representative of three independent experiments). The white arrows indicate accumulation in the nuclei of macrophages. Scale bars, 100 μm. **f** Lactate secretion and **g** production of IL-1β by BMDMs from C57BL/6 and HIF-1[c] mice left untreated (Ctrl) or infected with the Δ_ku80_ or Δ_pksP_ strains for 24 h ($n = 4$). **h** Levels of total and p-p70S6K in macrophages left untreated (Ctrl) or infected with the Δ_ku80_, Δ_pksP_, Δ_rodA/pksP_, or melanin-coated Δ_rodA/pksP_ strains for 3 h (representative of three independent experiments, with β-actin used as loading control). The pixel density of the p-p70S6K/p70S6K ratio was normalized to β-actin. Data are expressed as mean values ± SEM. _P_-values were calculated using Student's two-tailed _t_ test or one-way ANOVA with Tukey's multiple comparisons test.

ability of macrophages to reprogram their metabolism, leading to lower levels of secreted lactate. Along the same line, blocking mTOR (Fig. 4d) or Akt (Supplementary Fig. 4d) impaired the production of cytokines after infection.

Based on evidence that mTOR-mediated induction of glycolysis requires activation of HIF-1α and stimulation of glycolytic enzymes[42], we next assessed the expression of HIF-1α in infected macrophages. We found higher amounts of HIF-1α accumulated

in the nucleus of cells infected with the Δ*ku80* than the Δ*pksP* strain (Fig. 4e; Supplementary Fig. 4e), a finding indicating a melanin-dependent induction of HIF-1α-mediated gene expression. Likewise, HIF-1α was more abundant in lysates from macrophages infected with Δ*ku80* conidia (Supplementary Fig. 4f), and a similar profile was determined for *HIF1A* mRNA (Supplementary Fig. 4g). To further investigate the link between activation of HIF-1α and the metabolic reprogramming of macrophages, we assessed the impact of HIF-1α deficiency in response to infection using BMDMs from wild-type (C57BL/6) and myeloid-restricted HIF-1α-deficient (HIF-1[c]) mice. In contrast to wild-type BMDMs, the ability of melanin to induce lactate secretion was lost in HIF-1[c] cells (Fig. 4f), a finding recapitulated in human macrophages upon treatment with ascorbate, a co-factor for the hydrolases that negatively regulate HIF-1α (Supplementary Fig. 4h). Production of the proinflammatory cytokine IL-1β, known to be transcriptionally regulated by HIF-1α[9], was also impaired in HIF-1[c] BMDMs (Fig. 4g), a finding in line with the lower levels of cytokines produced by human macrophages treated with ascorbate (Supplementary Fig. 4i). In support of the link between fungal melanin and mTOR/HIF-1α signaling, infection of macrophages with melanin-coated live Δ*rodA/pksP* conidia restored both p-p70S6K (Fig. 4h) and HIF-1α translocation to the nucleus (Fig. 4e) to levels comparable with those obtained after infection with the Δ*ku80* strain. Taken together, these results suggest that signaling via mTOR and HIF-1α is required for the melanin-mediated activation of glycolysis in macrophages.

**Remodeling of calcium signaling enables host glycolysis.** Since the ability of fungal melanin to sequester calcium inside the phagosome is a major inhibitory mechanism of intracellular signaling pathways[43], we evaluated the contribution of calcium signaling to macrophage metabolism during infection. In line with the interference of fungal melanin with calcium responses, live imaging of macrophages preloaded with the calcium indicator Fluo-4-AM revealed that infection with Δ*pksP* or Δ*rodA/pksP*, but not Δ*ku80* conidia, triggered a sustained accumulation of cytosolic calcium early after infection (Fig. 5a, b; Supplementary Fig. 5a). Importantly, stimulation of macrophages with live Δ*rodA/pksP* conidia coated with purified melanin abrogated cytosolic calcium flux in a similar fashion to the Δ*ku80* strain, confirming melanin as a master regulator of calcium-mediated responses. This effect was dependent on fungal viability since stimulation with inactivated Δ*rodA/pksP* conidia or melanin ghosts instead failed to promote cytosolic calcium accumulation (Supplementary Fig. 5b).

To ascertain the specific contribution of melanin-mediated regulation of calcium signaling to host cellular metabolism, we next performed calcium depletion and signaling inhibition during infection and evaluated the induction of glycolysis. Strikingly, the depletion of extracellular calcium sources had no effect on calcium signaling and metabolism after infection with Δ*pksP* conidia (Fig. 5c). In contrast, incubation with the cell-permeable EGTA-AM further inhibited calcium responses to Δ*pksP* conidia, suggesting that intracellular calcium sources regulate the metabolic reprogramming. Because calcium concentrations in different subcellular compartments exert distinct functional effects[44], we performed the depletion of endoplasmic reticulum (ER) calcium stores with thapsigargin, a sarco/ER calcium ATPase (SERCA) inhibitor, during infection. Upon inhibiting SERCA, the impaired ability of macrophages to secrete lactate in response to infection with Δ*pksP* conidia was rescued (Fig. 5d). To further differentiate store depletion versus calcium entry, we evaluated lactate secretion following calcium supplementation. In these

conditions, cells infected with the Δ*pksP* conidia still failed to upregulate lactate levels (Fig. 5e). In addition, we silenced stromal-interacting molecule 1 (STIM1), a calcium sensor essential for calcium store depletion-triggered calcium influx[45] (Supplementary Fig. 5c). The inhibition of STIM1 also did not alter the profile of lactate secretion by macrophages infected with the Δ*ku80* or Δ*pksP* strains (Fig. 5f). Together, these observations indicate that ER calcium stores, independently of calcium entry, regulate intracellular signaling pathways, which are required for the activation of glucose metabolism by fungal melanin.

To gain further insight into the calcium-induced signaling pathways mediating activation of glycolysis by melanin, we next tested whether calcium/calmodulin (CaM) signaling was involved. We treated macrophages with W7, a specific CaM antagonist, and assessed the levels of lactate secretion after infection. In these conditions, macrophages infected with the Δ*pksP* strain displayed increased levels of secreted lactate (Fig. 5g), a finding in accordance with the upregulation of glycolytic genes (Supplementary Fig. 5d), and indicating that the inhibition of calcium/CaM signaling by melanin is required for the transcriptional induction of glycolysis. To further illustrate the regulation of immunometabolic signaling by fungal melanin, we next evaluated the influence of calcium signaling on mTOR expression. In line with the impaired phosphorylation of the mTOR target p70S6 kinase (Fig. 4b), mTOR expression was also decreased during infection with Δ*pksP* conidia, a defect that was rescued by the CaM antagonist W7 (Supplementary Fig. 5e). Because melanin blocks CaM recruitment to conidia-containing phagosomes[43], we next assessed mTOR recruitment to the phagosome. Macrophages infected with the Δ*pksP* strain displayed a lower percentage of mTOR[+] phagosomes when compared with cells infected with Δ*ku80* conidia (Fig. 5h, i). Importantly, however, mTOR recruitment was restored following CaM inhibition, a finding implying a direct link between melanin-mediated restraining of calcium signaling and activation of mTOR and the downstream metabolic reprogramming of macrophages. Furthermore, macrophages infected with Δ*pksP* conidia and treated with either thapsigargin or W7 displayed an enhanced ability to produce cytokines (Fig. 5j). Collectively, these results highlight a mechanism whereby the inhibition of calcium/CaM signaling enables efficient immunometabolic responses.

## Discussion

The metabolic reprogramming of innate immune cells is required to generate protective inflammation and drive antimicrobial defenses[12,20,46]. We establish fungal melanin as an essential PAMP required for the induction of glycolysis in macrophages and the resulting antifungal immune responses. DHN–melanin is a major determinant of the fungal interaction with the innate immune system[35], endowing *A. fumigatus* with the ability to survive killing by phagocytes, namely by blocking phagosome biogenesis[43,47] and acidification of phagolysosomes[48], and preventing phagocyte apoptosis[49]. We now show that the host has in turn evolved strategies to counter the inhibitory mechanisms deployed by melanin, by sensing its removal during intracellular swelling or cell wall remodeling inside the phagosome and using these signals to rewire cellular metabolism and promote antifungal immune responses.

The innate immune system is equipped to respond to the expression of virulence factors from fungi[16]. The identification of the MelLec receptor, which recognizes DHN–melanin and activates antifungal defenses, highlights the evolution of the receptor repertoire in innate cells towards the efficient recognition of fungal melanin[32]. Because the induction of glycolysis by melanin, but not MelLec expression, is evolutionarily conserved between

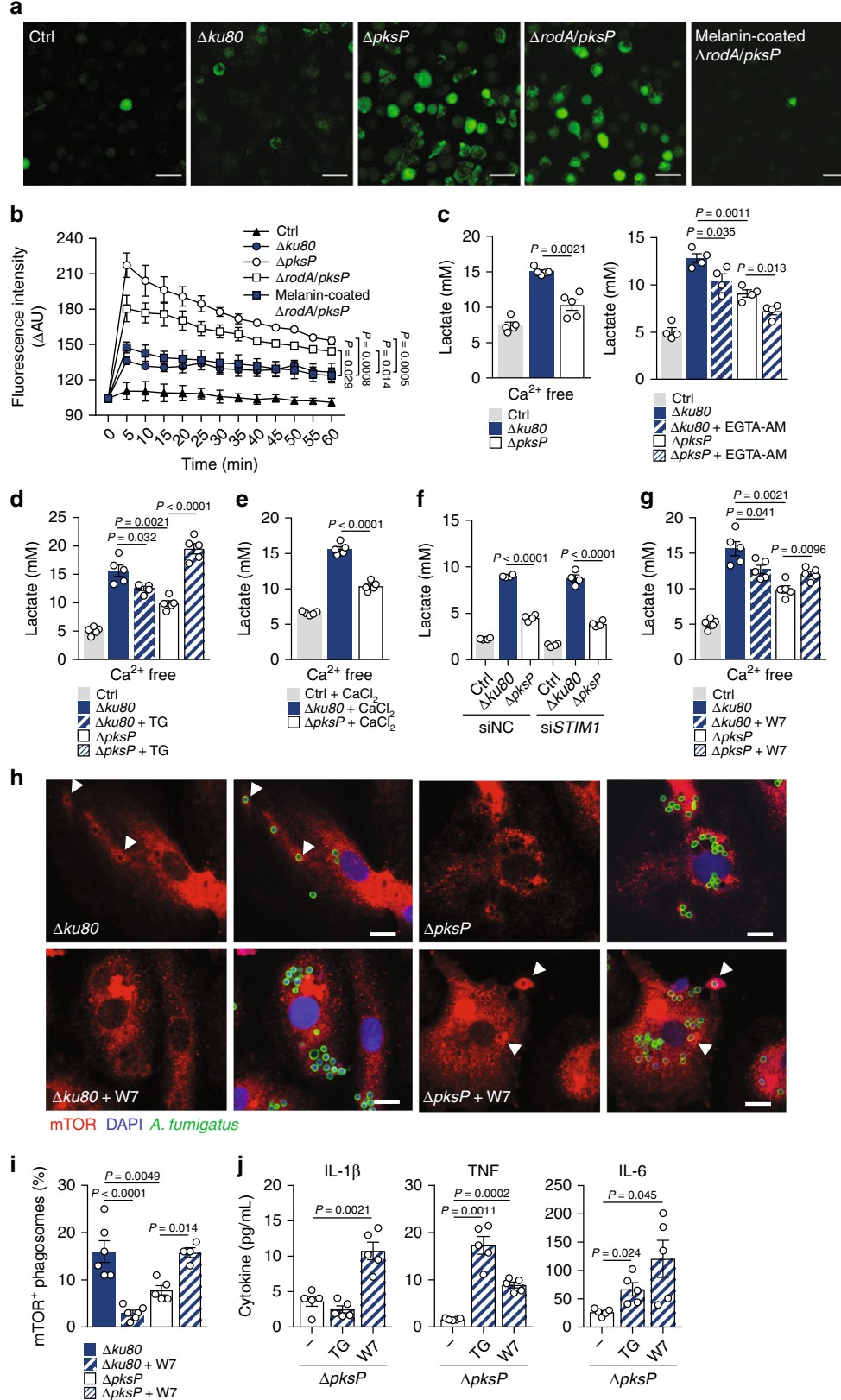

**h** mTOR  DAPI  *A. fumigatus*

human and murine macrophages, MelLec appears to be largely redundant for their metabolic reprogramming. Instead, the germination associated with the active removal of melanin in live fungi within the phagosome is required for host cells to reorient their metabolism towards protection. As such, melanin can be considered a vita-PAMP, molecules expressed by viable pathogens that are sensed by the host to weigh the intensity of specific antimicrobial responses[50]. It is noteworthy that the bacterial vita-PAMP cyclic-di-adenosine monophosphate triggers stress-mediated autophagy to restrain phagocyte death[51], and these molecular pathways are also well-known targets for the action of melanin during infection[47]. How is the outer layer of the

**Fig. 5 Calcium signaling regulates glucose metabolism in response to *A. fumigatus*. a** Micrographs of macrophages preloaded with Fluo-4-AM and left untreated (Ctrl) or infected with the Δ*ku80*, Δ*pksP*, Δ*rodA/pksP*, or melanin-coated Δ*rodA/pksP* strains for 15 min (representative of three independent experiments). Scale bars, 35 μm. **b** Quantification of Fluo-4-AM fluorescence in 5 min intervals for 1 h, and expressed as the difference in arbitrary units (ΔAU) between infected and uninfected cells. **c** Lactate secretion by macrophages left untreated (Ctrl) or infected with the Δ*ku80* or Δ*pksP* strains for 24 h using calcium-free medium ($n = 5$) or 500 μM EGTA-AM ($n = 4$). **d** Lactate secretion by macrophages left untreated (Ctrl) or infected in calcium-free medium for 24 h and in the presence of 2 μM thapsigargin (TG) ($n = 5$) or **e** 2 mM CaCl$_2$ ($n = 5$). **f** Lactate secretion by macrophages left untreated (Ctrl) or infected and silenced with a *STIM1* siRNA (si*STIM1*) ($n = 4$). A scrambled siRNA was used as negative control (siNC). **g** Lactate secretion by macrophages left untreated (Ctrl) or infected for 24 h in the presence of 25 μM W7 ($n = 5$). **h** Immunofluorescent staining for mTOR and **i** quantification of mTOR$^+$ phagosomes in infected macrophages, without or with 25 μM W7 for 2 h (representative of two independent experiments). The white arrows indicate the recruitment of mTOR to the phagosome. Scale bars, 10 μm. Data on quantification were determined by analyzing at least 200 phagosomes. **j** Production of IL-1β, TNF, and IL-6 by macrophages infected with the Δ*pksP* strain for 24 h without or with 2 μM TG or 25 μM W7 ($n = 5$). Data are expressed as mean values ± SEM. *P*-values were calculated using Student's two-tailed *t* test or one-way ANOVA and two-way ANOVA with Tukey's multiple comparisons test.

cell wall remodeled to shed melanin during germination and which signals are generated remains unclear. Could there be an intracellular receptor other than MelLec involved in melanin recognition? It is possible, since fungal ligands, namely β-(1,3)-glucan, directly activate the inflammasome upon their release during infection[52], and melanin could behave analogously. In fact, both the metabolic reprogramming of macrophages and inflammasome activation[53] rely exclusively on infection by live conidia, and melanin particles have been occasionally detected in the cytosol of infected cells (Chamilos, unpublished), raising the intriguing possibility that inflammasome activation and induction of glycolysis in macrophages are molecular events coupled via melanin-mediated signaling.

The recognition of β-1,3-glucan from the cell wall is required for the metabolic reprogramming of myeloid cells in response to *C. albicans*[11,20]. Surface exposure of β-1,3-glucan on *A. fumigatus* appears instead largely redundant for macrophage glycolysis given its broad expression on the surface of albino conidia[47]. Our data nevertheless suggest a minor contribution from signals other than melanin. Whether these include other cell wall polysaccharides or fungal-derived products requires further investigation. The differential activation of glucose metabolism upon exposure of host cells to distinct fungal PAMPs also reflects important differences in the outcome of the host–fungus interaction. In certain cases, fungi can hijack nutritional weaknesses of host cells that are a direct consequence of immunometabolic shifts to their own benefit; the terminal commitment of macrophages to glycolysis after infection with *C. albicans* is subverted by the efficient shift of fungal metabolism to rapidly deplete glucose and trigger cell death[22]. Because of lacking nutrients in the lung microenvironment, *A. fumigatus* resorts to different strategies to survive, which include the cross-pathway control system and degradation of proteins to acquire amino acids and the glyoxylate cycle to produce carbohydrates from lipids available at the site of infection[54]. In contrast to *C. albicans*, the metabolic reprogramming of macrophages in response to *A. fumigatus* represents instead an advantageous mechanism of host defense, highlighting the immunometabolic repurposing of immune cells toward enhanced glycolysis as an attractive therapeutic strategy in aspergillosis.

Despite the remarkable differences in the nature of the fungal ligands and the mechanisms through which these deliver signals required to induce glycolysis, the pathways that are activated in response to different fungal pathogens appear to converge on a common signaling axis involving mTOR and HIF-1α. This pathway represents a major signaling hub that regulates metabolic changes underlying trained immunity in myeloid cells in response to β-1,3-glucan[11,17–19]. Although trained immunity induced by melanin has not been explored, we provide evidence that this pigment is endowed with the ability to regulate mTOR

and HIF-1α signaling. Our findings are supported by proteomics of conidia-containing phagolysosomes[55], in which melanin was revealed as a major regulator of mTOR activator/regulator (LAMTOR), a member of the Ragulator/LAMTOR complex known to regulate mTOR[56]. Importantly, mTOR suppresses factor inhibiting HIF-1 (FIH-1), a negative regulator of HIF-1α[57], and FIH-1 may represent one central node congregating mTOR and HIF-1α signaling towards the activation of glycolysis in response to infection. Myeloid HIF-1α has been shown to be required for protection against aspergillosis, a phenotype attributed to impaired chemokine production and enhanced neutrophil apoptosis, resulting in a net decrease of neutrophil numbers[58]. Because HIF-1α also regulated cytokine production and the metabolic activity of human dendritic cells infected with *A. fumigatus*[59], it may deliver broader implications to antifungal immunity, with consequences to different cell types and involving distinct effector mechanisms.

What are the molecular signals that bridge melanin removal and the mTOR/HIF-1α axis culminating with the metabolic reprogramming of macrophages? We demonstrate that the immunometabolic reprogramming induced by *A. fumigatus* depends on calcium sequestration inside the phagosome by melanin which in turn triggers glycolysis-promoting signals. Calcium-mediated regulation of glycolysis has been known for several decades; for example, calcium overload was observed concomitantly to glycolytic inhibition in heart tissue[60], whereas calcium depletion promoted increased levels of glycolytic intermediates in cerebral cortex slices[61]. We demonstrate that calcium/CaM signaling orchestrates mTOR recruitment to the phagosome, an effect directly regulated by fungal melanin. Phagosome and entotic vacuole fission depend on mTOR localization to vacuolar membranes surrounding engulfed cells[62] and, likewise, we propose that the subcellular redistribution of mTOR to melanin-containing phagosomes is required for its activation. Because the spatial organization of mTOR is coordinated through a variety of sensors and regulators that converge on Rag GTPases[63], the phagosomal regulation of LAMTOR by melanin[55] suggests similar mechanisms during fungal infection, in line with the ability of melanized conidia to inhibit phagosomal recruitment of CaM, and prevent the specialized autophagy pathway LC3-associated phagocytosis (LAP)[43]. By deploying melanin, the fungus promotes calcium sequestration inside the phagosome and inhibits the recruitment of LAPosome machinery components, including the NADPH oxidase and CaM, to block LAP and avoid elimination[43,47]. Our data demonstrate that melanin, by blocking the activation of LAP, contributes to fungal persistence within the phagosome, allowing conidial germination and removal of melanin, providing the necessary signals to redirect macrophage metabolism towards glycolysis and efficient fungal clearance.

Our study provides mechanistic insights into the interplay between *A. fumigatus* and the glucose metabolism of immune cells. It has, however, limitations, such as the likely involvement of metabolic pathways other than glycolysis in the activation of antifungal immunity. Cellular metabolism consists of highly interconnected pathways that feed into each other and it is challenging, if not at all impossible, to analyze them in an entirely independent way. Nevertheless, our results suggest a dominant role for glycolysis and represent a promising first step toward the elucidation of the metabolic profiles involved in the activation of antifungal immunity. Current therapeutic limitations and concerns over the emergence of antifungal resistance are inspiring the search for novel host-directed therapies. Understanding how metabolic networks coordinate immune cell function may lead to innovative therapeutic approaches or metabolic adjuncts to reorient host cells towards immune protection against infection.

## Methods

**Ethics statement**. The functional experiments involving cells isolated from the peripheral blood of healthy volunteers at Hospital of Braga, Portugal, were approved by the Ethics Subcommittee for Life and Health Sciences (SECVS) of the University of Minho, Portugal (no. 014/015). Experiments were conducted according to the principles expressed in the Declaration of Helsinki, and participants provided written informed consent.

**Mice**. Eight-week-old gender- and age-matched C57BL/6 mice were bred under specific pathogen-free condition and kept at the Life and Health Sciences Research Institute (ICVS) Animal Facility. Mice were fed ad libitum and kept under light/dark cycles of 12 h, temperature of 18–25 °C and humidity of 40–60%. Animal experimentation was performed following biosafety level 2 (BSL-2) protocols approved by the Institutional Animal Care and Use Committee (IACUC) of University of Minho, and ethical and regulatory approvals were consented by SECVS (no. 074/016). All procedures in vivo followed the EU-adopted regulations (Directive 2010/63/EU), and were conducted according to the guidelines sanctioned by the Portuguese ethics committee for animal experimentation, Direção-Geral de Alimentação e Veterinária (DGAV).

***Aspergillus* strains and culture conditions**. The A1163 Δ*ku80*[64] and B-5233[65] strains of *A. fumigatus* were used as wild-type strains. The Δ*rodA*, Δ*pksP*, and Δ*rodA/pksP* deletion mutants[43,47] and the mutant strains in the DHN–melanin biosynthetic pathway (Δ*pksP*, Δ*ayg1*, Δ*abr1*, Δ*abr2*, Δ*arp1*, and Δ*arp2*) in the B-5233 background[66] were generated previously. Wild-type CBS110.46 and CBS386.75 strains with albino conidia[38] were also used, as indicated. All strains were grown on 2% malt extract agar or YAG agar for 7 days at 28 °C. The conidia were harvested from agar slants using phosphate buffer saline (PBS) (Gibco, Thermo Fisher Scientific) with 0.05% Tween 20 (Sigma-Aldrich), followed by gentle agitation and subsequent filtration through a 40-μm-pore size cell strainer (Falcon). The concentration of conidia/mL was determined by counting in a Neubauer chamber. Swollen conidia and germ tubes were obtained after incubation in Sabouraud liquid culture medium at 37 °C for 4 h and 6 h, respectively. Heat inactivation of conidia was performed by incubation for 30 min at 90 °C, whereas UV inactivation was performed by exposing conidia to UV light for 3 h.

**Extraction of *A. fumigatus* melanin and conidial coating**. Melanin from A1163 Δ*ku80* conidia was isolated using a combination of proteolytic (proteinase K; Sigma-Aldrich) and glycohydrolytic (Glucanex; Novo) enzymes, denaturant (guanidine thiocyanate) and hot, concentrated HCl (6 M) to treat conidia, resulting in an electron-dense layer similar in size and shape to the original conidial melanin layer but without the underlying cell components, named melanin ghosts[67]. Melanin-coated Δ*rodA/pksP* conidia were obtained by overnight coating of conidia with different concentrations of melanin ghosts, previously sonicated with pulses of 5 s during 1 min, at room temperature (RT).

**Isolation of PBMCs and generation of MDMs**. Peripheral blood mononuclear cells (PBMCs) were enriched from buffy coats or whole blood by density gradient using Histopaque®-1077 (Sigma-Aldrich), washed twice in PBS and resuspended in RPMI-1640 culture medium with 2 mM glutamine (Gibco, Thermo Fisher Scientific) supplemented with 10% human serum (Sigma-Aldrich), 10 U/mL penicillin/streptomycin and 10 mM HEPES (Thermo Fisher Scientific) (cRPMI). Monocytes were isolated from PBMCs using positive magnetic bead separation with anti-CD14+-coated beads (MACS Miltenyi) according to the manufacturer's instructions. Isolated monocytes were resuspended in cRPMI medium, and seeded at a concentration of $1 \times 10^6$ cells/mL in 24-well and 96-well plates (Corning Inc.) and eight-well chamber slides (LAB-TEK, Thermo Fisher Scientific) for 7 days in the presence of 20 ng/mL recombinant human granulocyte macrophage colony-stimulating factor (GM-CSF, Miltenyi Biotec) or 20 ng/mL of macrophage colony-stimulating factor (M-CSF, Miltenyi Biotec). The culture medium was replaced every 3 days, and acquisition of macrophage morphology was confirmed by visualization in a BX61 microscope (Olympus).

**Generation of BMDMs**. Bone marrow-derived macrophages (BMDMs) were obtained from femur and tibia bones of male or female 8-week-old C57BL/6 or Hif1a^fl/fl^-LysMcre^+/+^, hereafter referred as HIF-1^c^, mice. Briefly, bone marrow was harvested and cultured in Dulbecco's modified medium (DMEM) with 1% penicillin/streptomycin, 1% L-glutamine, and 10% FBS (Gibco, Thermo Fisher Scientific), supplemented with 20 ng/mL M-CSF (Peprotech) for 7 days at 37 °C and 5% $CO_2$, with the addition of 20 ng/mL of M-CSF at day 4 of differentiation. Acquisition of macrophage morphology was confirmed by visualization in a BX61 microscope (Olympus).

**Cell stimulations and treatments**. Unless otherwise indicated, MDMs or BMDMs ($5 \times 10^5$/well in 24-well plates) were infected with *A. fumigatus* conidia at a 1:2 or 1:10 effector-to-target ratio or stimulated with 100 μg/mL of melanin ghosts or with 50, 75, 100, and 200 μg/mL 1,8-DHN for 24 h at 37 °C and 5% $CO_2$. For experiments involving glucose depletion, RPMI-1640 medium without glucose (Thermo Fisher Scientific) was used. To interfere with cellular metabolism, MDMs were pretreated for 1 hr with 5, 10, or 20 mM 2-DG, 30 μM 3PO, 500 nM 6-AN, 10 μM wortmannin, or 10 nM rapamycin, and for 3 h with 50 μM (+)-sodium L-ascorbate. For experiments involving calcium manipulation, MDMs were pre-treated for 30 min with 2 μM thapsigargin, or treated for 10 min with 25 μM W7 (all from Sigma-Aldrich) in calcium-free DMEM (Thermo Fisher Scientific) or for 1 hr with 500 μM EGTA-AM (Thermo Fisher Scientific) after infection. In some conditions, MDMs were cultured for 1 h with calcium-free DMEM followed by the addition of 2 mM $CaCl_2$ during infection. To assess cell viability, 4 h prior to the end of infection, 50 μL of alamarBlue™ Cell Viability Reagent (Thermo Fisher Scientific) were added to each well. Viability was assessed by the quantification of relative fluorescence units (RFU) using a Varioskan Flash fluorescent plate reader (Thermo Fisher Scientific) with a fluorescence excitation wavelength of 570 nm and an emission wavelength of 600 nm. In some experiments, viability was evaluated by flow cytometry using annexin-V/propidium iodide 6 h after infection. In all experiments involving MDMs, the data were assessed in triplicates, and are shown as the mean value for each individual.

**siRNA-mediated gene silencing**. MDMs ($5 \times 10^4$/well in 96-well plates) were incubated for 72 h at 37 °C and 5% $CO_2$ in Accell Delivery Media in the presence of either 1 μM STIM1 siRNA or a non-targeting siRNA control (siNC) (Dharmacon). After incubation, the transfection media was removed, and cells were infected with *A. fumigatus* conidia at a 1:10 effector-to-target ratio. Pooled replicates from three different individuals were collected after 24 h to measure lactate secretion and cytokine production. The mRNA knockdown was confirmed by qPCR.

**RNA sequencing**. MDMs ($5 \times 10^5$/well in 24-well plates) were infected with *A. fumigatus* conidia at a 1:2 effector-to-target ratio, and pooled replicates from three different individuals were collected after 2 and 6 h. Uninfected MDMs were cultured in parallel as controls. Sample processing, sequencing, and analysis were performed at IMGM Laboratories GmbH (Germany). Briefly, the total RNA was isolated using the RNeasy Mini Kit (Qiagen) according to the manufacturer's instructions, including on-column DNase digestion. The total RNA was eluted in 30 μL of RNase-free water. The quality of the total RNA was analyzed with the 2100 Bioanalyzer using RNA 6000 Nano and Pico LabChip kits (Agilent Technologies). Library preparation was performed using the TruSeq® Stranded mRNA HT technology, according to the manufacturer's protocol. All single libraries were pooled into a final sequencing library with an equal DNA amount per sample. The final sequencing library generated by pooling was quantified using the highly sensitive fluorescent dye-based Qubit® dsDNA HS Assay kit (Invitrogen) before sequencing at a final concentration of 1.8 pM and with a 1% PhiX v3 control library spike-in (Illumina) on the NextSeq500 sequencing system (Illumina). For the clustering and sequencing of samples, a high-output single-end 75 cycles (1 × 75 bp SE) run was performed under the control of the NextSeq Control Software (NCS, Illumina). Quality control was carried out using NCS and Real Time Analysis 2.4.11 softwares applying the *FastQ only* pipeline. Read data were imported into the CLC Genomics Workbench (CLC bio, Qiagen), and reads were mapped against the human reference genome (GRCh37.p13) with subsequent counting and distribution of reads across genes and transcripts. The expression values were then processed to reads per kilobase million (RPKM), a normalized measure of relative abundance of transcripts[68], followed by analysis using the EdgeR Bioconductor package[69] to identify differentially expressed genes with a fold change value ≥2 or ≤ −2 with a false discovery rate (FDR)-corrected *p*-value < 0.05. Heatmaps were created for the most significantly represented genes of a specific functional class using the Morpheus tool (Broad Institute; https://software.broadinstitute.org/morpheus/). For pathway analysis, the annotated hallmark gene sets from the Molecular Signatures Database (MSigDB)[33] were used and enrichment analysis was performed using the Gene Ontology Biological Processes category in the Gene Ontology Consortium software (http://www.geneontology.org/).

**Quantification of glucose and lactate by HPLC**. After infection, supernatants were removed, centrifuged, and transferred to HPLC tubes. Glucose and lactate levels were determined using a Gilson pump system (Gilson) with a 54 °C HyperREZ XP Carbohydrate H$^+$ 8 µM (Thermo Fisher Scientific) column and a refractive index detector (IOTA 2, Reagents). The mobile phase consisting of 0.0025 M $H_2SO_4$ was filtered and degasified for at least 45 min before use. Standard solutions were prepared in Milli-Q water (Millipore). All data were analyzed using the Gilson Uniprot Software, version 5.11.

**LC–MS/MS-targeted metabolomics**. MDMs ($2 \times 10^6$/well, in six-well plates) were infected with *A. fumigatus* conidia at a 1:2 effector-to-target ratio for 6 h at 37 °C in 5% $CO_2$. To detach the cells, the bottom of the well was scraped lightly with a cell scraper. The resulting cell suspensions, obtained from pooled replicates from three different individuals, were centrifuged at 4 °C, the supernatant was discarded, and the resulting pellet was immediately frozen in liquid nitrogen to quickly quench the metabolism. To maximize overall metabolite yield, 280 µL of MeOH:MTBE (Methyl tert-butyl ether) (4:1, v/v) was added to the pellet, followed by three cycles of freezing/thawing in liquid nitrogen and in cold-water bath, respectively, for 10 s each. A sonication at 15 W for 6 min was then performed, followed by 1 min of vortexing and a centrifugation at 4 °C. From the resulting supernatant, a volume of 250 µL was carefully transferred to a new Eppendorf tube. In addition, 280 µL of MeOH:$H_2O$ (4:1, v/v) was added to the pellet, and the same procedure above was performed. At the end, 250 µL of the resulting supernatant was mixed with the first 250 µL of supernatant. The samples were stored at −80 °C until further analysis. To perform the metabolomics analysis, standards of analytical grade (Sigma-Aldrich) were used for external calibration and adjusted to the corresponding levels in the samples. Ultrapure water, purified using a Milli-Q system (Millipore), was used for buffer preparation. Liquid chromatography–mass spectrometry (LC–MS)- grade methanol and acetonitrile (both from Thermo Fisher Scientific) were used. Tributylamine (≥99.5%) (Sigma-Aldrich) stock solutions of all the standards were prepared at 1000 ppm in water, and stored at −20 °C. The LC–MS/MS analyses were performed in an Agilent 1290 Infinity (Agilent Technologies) using a 1290 Infinity Binary Pump (1200 bar) and a 1260 Quaternary Pump (400 bar). The LC system was coupled to an Agilent 6460 triple-quadrupole mass spectrometer using an electrospray ionization (ESI) interface working in multiple reaction monitoring (MRM) mode. The chromatographic method, the MS parameters and the setup arrangement were based on that described by Agilent Technologies, Inc, with minor modifications. Briefly, this method uses tributylamine as an ion-pairing reagent, with buffer A composed of 97% water and 3% methanol, 10 mM tributylamine, 15 mM glacial acetic acid (VWR), and buffer B composed of 10 mM tributylamine, 15 mM glacial acetic acid, prepared in methanol. The transitions showing the highest signal to noise ratios were used for the quantification of the analytes in samples.

**Mouse infection**. For the in vivo studies, at day 3 before infection and until the end of experimental protocol, 100 mg/Kg of 2-DG was administered to mice daily via intraperitoneal route (i.p.). Control groups consisted of mice to which an equal volume of vehicle (sterile PBS) was administered. At day 0, mice were challenged with $1 \times 10^8$ live conidia of *A. fumigatus* (A1163 Δku80 strain) using a noninvasive intranasal (i.n.) infection procedure upon anesthesia with 75 mg/Kg of ketamine (Ketamidor®, Ritcher Pharma) and 1 mg/Kg of medetomidine (Domtor®, Ecuphar). At 24 h post infection, mice were killed, and the lungs were PBS-perfused and excised, excluding the trachea and major bronchi. For assessment of fungal burden, lung single-cell suspensions were serially diluted and plated on solid growth media.

**Determination of systemic glucose**. On the day of the experiment, mice were starved for 3–4 h in the morning before measurements of blood glucose levels at 0 h, by snipping the very end of the tail to collect a drop of blood in a Glucocard®+ (Arkray), which was read using the ACCU-CHEK Performa glucometer (Roche). Afterwards, mice were infected intranasally with $1 \times 10^8$ conidia of the Δku80 strain with ad libitum access to food. After 20 h of infection, mice were again starved for 3–4 h before glucose measurement in the blood. The systemic glucose concentration was calculated as the glucose (mM) per gram of mice body weight.

**Isolation of splenocytes**. Infected animals were sacrificed at day 1 post infection with posterior spleen excision. The excised spleen was minced into small fragments with a plunger end of a syringe and forced through a 70-µm cell strainer (Corning Inc.) Upon washing the cells with cold sterile PBS, the resulting cell pellet was resuspended in 2 mL of pre-warmed ACK lysis buffer (0.15 M $NH_4Cl$, 10 Mm $KHCO_3$, and 0.1 mM EDTA). Afterwards, the cells were centrifuged at 1600 rpm for 5 min at RT. Finally, the cells were counted and adjusted to a final concentration of $5 \times 10^6$ cells/mL of pre-warmed RPMI (Gibco, Thermo Fisher Scientific) supplemented with 10% FBS (Gibco, Thermo Fisher Scientific) and 200 µL of the cellular suspension were seeded in round bottom 96-well plates (Corning Inc.).

**FACS analysis and sorting**. After infection, the lungs were excised and collected in incomplete Dulbecco's Modified Eagle Medium (iDMEM) culture medium (Gibco, Thermo Fisher Scientific). Perfused lungs were chopped in small fragments

and digested at 37 °C for 30 min in iDMEM culture medium containing 1 mg/mL of collagenase D (Sigma-Aldrich). Subsequently, the tissue was forced through a 70-µm cell strainer, and contaminating red blood cells were lysed. Leukocytes were isolated by Percoll (GE Healthcare Bio-Sciences Ab) density gradient, and finally resuspended in FACS buffer (PBS containing 2% FBS and 2 mM EDTA). To assess cell viability, cells were stained for 30 min in the dark with the Zombie Violet fluorescent dye (BioLegend) and resuspended in FACS buffer. For surface marker staining, cell suspensions were stained for 30 min on ice while protected from light with the indicated antibodies. Pellets were washed and resuspended in fresh FACS buffer prior to analysis. Gating for myeloid subpopulations in the lung was performed to analyze the composition of lung-infiltrating cells using a combination of the following antibodies: BV510 anti-mouse CD45 (clone 30-F11), BV605 anti-mouse CD11c (clone N418), PE-Cy7 anti-mouse CD11b (clone M1/70), and APC anti-mouse F4/80 (clone BM8) (all from BioLegend)[22]. Data were obtained on a BD FACS LSRII instrument (Becton Dickinson), and processed using FlowJo (Tree Star Inc). For the sorting of macrophage populations in the lung, CD45 + positive cells were isolated using positive magnetic bead separation with anti-CD45$^+$ coated beads (MACS Miltenyi) before surface marker staining. Macrophages were sorted using a combination of the following antibodies: PerCP/Cy5.5 anti-mouse CD11c (clone N418), APC anti-mouse CD11b (clone M1/70), PE anti-mouse Siglec-F (clone E50-2440), APC-Cy7 anti-mouse Ly-6G (clone 1A8), FITC anti-mouse CD64 (clone X54-5/7.1), and PE/Cy7 anti-mouse CD45 (clone 30-F11) (from BioLegend or BD Biosciences)[70]. Data were obtained on a BD FACSAria II instrument and analyzed with the FACSDiva software (Becton Dickinson).

**Measurement of ADP/ATP ratio**. MDMs ($1 \times 10^5$/well in 96-well plates) were infected with *A. fumigatus* conidia for 6 h at a 1:10 effector-to-target ratio at 37 °C in 5% $CO_2$. After infection, the ADP/ATP ratio was measured using an assay kit (Sigma-Aldrich), according to the manufacturer's instructions. Briefly, culture medium was removed, and the ATP reagent was added to each well. Plate was incubated for 1 min at RT, and luminescence was read to obtain ATP levels. After 10 min of incubation at RT, the ADP reagent was added to each well, and luminescence was read. The ADP/ATP ratio was calculated by subtracting background values from ADP values and dividing the result by ATP values. Luminescence was read in a Fluoroskan FL Microplate Luminometer (Thermo Fisher Scientific).

**Phagocytosis assay**. To determine phagocytosis, MDMs ($5 \times 10^5$/well in 24-well plates) were infected with fluorescein isothiocyanate (FITC)-labeled conidia of *A. fumigatus* at a 1:5 effector-to-target ratio. The infection was synchronized for 30 min at 4 °C, and phagocytosis was initiated by shifting the co-incubation to 37 °C at 5% $CO_2$ for 1 h. Phagocytosis was stopped by washing wells with PBS, and extracellular conidia were stained with 0.25 mg/mL Calcofluor White (Sigma-Aldrich) for 15 min at 4 °C to avoid further ingestion. Wells were then washed twice with PBS, and cells were fixed with 3.7% (v/v) formaldehyde/PBS for 15 min. The number of MDMs with ingested green conidia was enumerated by examining the slides by fluorescence microscopy (Olympus), and data were expressed as percentage of MDMs that internalized one or more conidia.

**Conidiacidal activity assay**. Following differentiation, MDMs ($1 \times 10^5$/well in 96-well plates) were washed twice with cRPMI medium, and a suspension of *A. fumigatus* conidia was added at a ratio of 10:1 effector-to-target ratio. The cells were then incubated for 1 h at 37 °C and 5% $CO_2$ to allow the internalization of conidia. Medium containing the non-ingested conidia was removed, and wells were washed twice with pre-warmed PBS. To measure the conidiacidal ability, MDMs were allowed to kill the ingested conidia for 2 h at 37 °C in 5% $CO_2$. To determine the conidiacidal activity of splenocytes, cells ($1 \times 10^6$/well in 96-well plates) were infected with *A. fumigatus* conidia at a 1:5 effector-to-target ratio for 2 h at 37 °C and 5% $CO_2$. After incubation, culture plates were snap frozen at −80 °C, and thawed at 37 °C to cause cell lysis and release of ingested conidia. Serial dilutions of cell lysates were plated on solid growth media and, following a 24 h incubation at 37 °C, the number of colony-forming units (CFUs) was enumerated, and the percentage of CFU inhibition was calculated.

**Measurement of ROS production**. MDMs ($1 \times 10^5$/well) were plated in 96-well TC-treated dark clear bottom plates (Sigma-Aldrich) and infected with live *A. fumigatus* conidia at a 1:3 effector-to-target ratio. Infections were synchronized by centrifugation at 1000 rpm for 5 min, and then 10 µM dihydrorhodamine 123 (DHR) (Thermo Fisher Scientific) was added to each well, and the production of reactive oxygen species (ROS) was measured for a 24-h period using the Varioskan Flash fluorescent plate reader (Thermo Fisher Scientific). Excitation was performed at a wavelength of 480 nm, and emission was measured at a wavelength of 528 nm.

**Cytokine measurements**. MDMs and BMDMs ($5 \times 10^5$/well in 24-well plates) and splenocytes ($1 \times 10^6$/well in 96-well plates) were infected with *A. fumigatus* conidia at a 1:10 effector-to-target ratio for 24 h at 37 °C and 5% $CO_2$. PBMCs ($5 \times 10^5$/well in 96-well plates) were infected with UV-inactivated conidia at a 1:4 effector-to-target ratio for 7 days at 37 °C and 5% $CO_2$. After infection, supernatants were collected and cytokine levels were quantified using ELISA MAX Deluxe Set kits (BioLegend), according to the manufacturer's instructions. Quantitative cytokine

measurements were also performed on the supernatants of lung single-cell suspensions 24 h post infection.

**RNA isolation and qRT-PCR**. The total RNA from MDMs ($5 \times 10^5$/well in 24-well plates) was isolated at different time points after infection with *A. fumigatus* at a 1:10 effector-to-target ratio using the PureLink™ RNA Mini Kit (Thermo Scientific) according to the manufacturer's instructions. The total RNA from lungs was extracted using the GRS Total RNA Kit-Tissue (Grisp) at different time points after infection, according to the manufacturer's instructions. The concentration and quality of total RNA in each sample were determined by spectrophotometry using the ND-100 UV-visible light spectrophotometer (NanoDrop). One microgram of the total RNA was retro-transcribed using the first-strand cDNA Synthesis Kit (Nzytech). Quantitative PCR was performed in an Applied Biosystems 7500 Fast qPCR system (Applied Biosystems, Thermo Fisher Scientific), using the PowerUp SYBR Green Master Mix (Applied Biosystems, Thermo Fisher Scientific). Data were analyzed using the 7500 Software v2.0.6 software (Applied Biosystems, Thermo Fisher Scientific). Amplification efficiencies were validated, and the expression levels of the transcripts were normalized using the *ACTB* (human), *Ubb* (mouse) and *18S* (*A. fumigatus*) genes.

**Western blot analysis**. Human MDMs ($5 \times 10^5$/well in 24-well plates) were infected with *A. fumigatus* conidia for the indicated time points at a 1:10 effector-to-target ratio at 37 °C in 5% $CO_2$. After infection, cells were lysed in RIPA buffer (50 mM Tris, 250 mM NaCl, 2 mM EDTA, 1% NP-40, 10% glycerol, pH 7.2, and a mixture of protease inhibitors [Roche Molecular Biochemicals]). Cell lysis was performed at 4 °C for 30 min (with shaking), and samples were then centrifuged. The protein content was determined using the Bradford dye-binding (Bio-Rad) method. Laemmli buffer (Bio-Rad) was added to 20 µg of protein, and samples were boiled and separated on a 12% SDS-PAGE gel, and transferred to nitrocellulose membranes (Bio-Rad). Western blotting was performed according to the manufacturer's instructions, using the following primary antibodies: rabbit anti-phospho-p70S6 Kinase, rabbit anti-p70S6 Kinase, rabbit anti-Akt, rabbit anti-phospho-Akt, rabbit anti-mTOR (all from Cell Signaling), mouse anti-β-actin (Abcam), all diluted 1:1000, and rabbit anti-HIF-1α antibody (Abcam) diluted 1:500. Secondary antibodies used were anti-rabbit and anti-mouse, both diluted to 1:5000. The blots were developed using chemiluminescence (SuperSignal™ West Femto Maximum Sensitivity Substrate; Thermo Fisher Scientific), and detected with ChemiDoc™ XRS system (Bio-Rad). Signal intensities and quantifications were determined with the ImageLab 4.1 analysis software (Bio-Rad).

**Live cell imaging**. To perform live imaging of calcium, MDMs were seeded in eight-well-chamber slides (LAB-TEK, Thermo Fisher Scientific) ($3 \times 10^5$/well) and loaded with 3 µM of the calcium indicator Fluo-4AM (Thermo Fisher Scientific) according to the manufacturer's protocol. Briefly, MDMs were placed in serum-free HBSS (without $Ca^{2+}$, $MgCl_2$, and phenol red) and loaded with Fluo-4AM for 30 min at 37 °C, 5% $CO_2$. Cells were then washed twice with PBS, and infections with the indicated fungal strains were performed in cRPMI medium. Live cell imaging was performed on an Olympus FV1000 Plus Confocal Microscope (60× Luc Plan FL 0.70 NA objective) at 37 °C with 5% $CO_2$. All analyses and processing were made using ImageJ software (Fiji). To visualize the temporal changes in calcium, raw sequences were processed, and mean pixel intensity at each frame was measured. The data were first plotted as fluorescence intensity versus time (Z profile) and subsequently converted to relative scale (ΔF/F baseline). To perform imaging of HIF-1α, MDMs were seeded in eight-well-chamber slides (LAB-TEK, Thermo Fisher Scientific) ($3 \times 10^5$/well) and infected for 2 h with the indicated FITC-labeled fungal strain at 37 °C in 5% $CO_2$. After washing twice with PBS, the cells were fixed with 3.7% (v/v) formaldehyde/PBS for 15 min, permeabilized with 0.3% Triton™ X-100 (Sigma-Aldrich) in PBS for 10 min, blocked with 4% BSA (Sigma-Aldrich) for 1 h, and incubated overnight with primary antibody rabbit anti-HIF-1α antibody (dilution 1:100, Abcam). MDMs were then washed twice with PBS and incubated with anti-Rabbit IgG (H + L) Cross-Adsorbed Secondary Antibody Alexa Fluor® 568 conjugate (dilution 1:1000, Thermo Fisher Scientific) for 1 h at RT. Nuclei were stained with DAPI (dilution 1:1000, Thermo Fisher Scientific) for 10 min. Images were captured at ×100 magnification on an Olympus FV1000 Plus Confocal microscope. HIF-1α quantification was performed using ImageJ (Fiji), and expressed as nuclear/total HIF-1α staining of at least 60 cells for each condition. For immunofluorescence imaging of mTOR on phagosomes, MDMs were seeded on coverslips pretreated with polylysine, fixed with 4% PFA for 15 min at RT following by 10 min of fixation with ice-cold methanol at −20 °C, washed twice with PBS, permeabilized using 0.1% saponin (Sigma-Aldrich) and blocked for 30 min PBS with 2% BSA. After incubation with anti-mTOR antibody (dilution 1:250; Cell Signaling) for 1 h, slides were washed twice in PBS–BSA and stained with the Alexa Fluor 555 secondary antibody (Molecular Probes), followed by DNA staining with 10 µM TOPRO-3 iodide (642/661; Invitrogen). After the washing steps, slides were mounted in Prolong Gold antifading media (Molecular Probes). Images were acquired using a laser-scanning spectral confocal microscope (TCS SP2; Leica), LCS Lite software (Leica), and a ×40 Apochromat 1.25 NA oil objective using identical gain settings. A low fluorescence immersion oil (11513859; Leica) was

used, and imaging was performed at RT. Unless otherwise stated, mean projections of image stacks were obtained using the LCS Lite software and processed with Adobe Photoshop CS2. Phagosomes surrounded by a rim of fluorescence of the indicated protein-marker were scored as positive[43,47]. At least 200 phagosomes were analyzed for each condition in two independent experiments.

**Statistical analysis**. The data were expressed as means ± SEM. Statistical significance of differences were determined by two-tailed Student's *t* test, one-way ANOVA, or two-way ANOVA with post hoc tests for multiple comparisons ($P < 0.05$ was considered statistically significant). Analyses were performed in GraphPad Prism software.

## Data availability

RNA-seq data have been deposited in Gene Expression Omnibus (GEO) with the accession code GSE128661. The Molecular Signatures Database version 7.1, a collection of annotated gene sets for use with GSEA software, was used in the study [https://www.gsea-msigdb.org/gsea/msigdb/index.jsp]. All other data, materials, and reagents are available on request from the corresponding author. The source data underlying Fig. 1d–h, Fig. 2, Fig. 3, Figs. 4b–d and 4f–h, Fig. 5, and Supplementary Fig. 1, Fig. 2, Fig. 3, Fig. 4, and Fig. 5c–e are provided as a Source Data file.

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

## Acknowledgements

This work was supported by the Northern Portugal Regional Operational Programme (NORTE 2020), under the Portugal 2020 Partnership Agreement, through the European Regional Development Fund (FEDER) (NORTE-01-0145-FEDER-000013), the Fundação para a Ciência e Tecnologia (FCT) (SFRH/BD/136814/2018 to S.M.G., SFRH/BD/141127/2018 to C.D.O., PD/BD/137680/2018 to D.A., IF/00474/2014 to N.S.O., IF/01390/2014 to E.T., IF/00959/2014 to S.C., IF/00021/2014 to R.S., PTDC/SAU-SER/29635/2017 and CEECIND/04601/2017 to C.C., and CEECIND/03628/2017 to A.C.), the Institut Mérieux (Mérieux Research Grant 2017 to C.C.), and the European Society of Clinical Microbiology and Infectious Diseases (ESCMID Research Grant 2017 to A.C.). M.G.N. was supported by a Spinoza grant of the Netherlands Organization for Scientific Research. A.A.B. was supported by the Deutsche Forschungsgemeinschaft Collaborative Research Center/Transregio TR124 FungiNet (project A1). G.D.B. was funded by the

Wellcome Trust (102705), the MRC Centre for Medical Mycology and the University of Aberdeen (MR/N006364/1).

## Author contributions

S.M.G. designed the study, performed experiments and data analysis, and wrote the paper. C.D.O., C.F.C., and I.M. performed the animal experiments, V.A. prepared fungal components, D.A. and G.C. performed confocal microscopy, C.S.R. performed HPLC measurements, C.B.M., J.G., and S.C. performed flow cytometry and cell-sorting experiments, and G.C. performed the live recording of calcium levels. N.S.O. analyzed RNA-seq data, and M.F.G. and C.B. performed targeted metabolomics. A.M. supervised the collection of samples from healthy donors. R.t.H., L.L., T.M., P.P., E.T., F.R., L.A.B.J., K.L., J.M., J.F.L., A.C.J., and R.S. analyzed the data and contributed to the paper. G.D.B., A.A.B., and G.C. contributed to the paper. F.L.v.d.V., M.G.N., J.P.L., C.C., and A.C. designed the study and wrote the paper.

## Competing interests

The authors declare no competing interests.
