## [Peer Review File · Nature Communications]

Reviewers' comments:

Reviewer #1 (Remarks to the Author):

This study seeks to make a link between the fungal cell wall and host cell metabolism, two major topics in fungal host-pathogen interactions. The authors show a link between fungal melanin, host calcium flux, mTOR and HIF1alpha and Warburg shift.

I thought the introduction was excellent.

Figure 1: How many human macrophage replicates were undertaken from how many individuals? Were the macrophages infected with resting or swollen conidia or hyphae and at what MOI? Figure 1H the loss of viability suggests a low MOI and resting conidia. This is corroborated by the down regulation of cytokines at 6 hours. are you sure these macrophages are actually infected at later time points? Likewise I assume 1x10⁸ inoculum was used for the animal experiments but how pathogenic is the Ku strain and why wasn't CEA10 used?

Figure 2: 2B and D seem to show that all the phagocytosed conidia are killed, presumably the non phagocytosed conidia survive. So the issue seems that it is not clear what level of productive and sustained macrophage infection there is in the model. How specific is 2DG for glycolysis?

Figure 3: The issue of resting vs swollen conidia is now addressed. This reveals that melanin induces glycolysis and lactate production. The genetics is convincing.

Figure 4: How many replicates were used for the transcriptomics from how many patients? Can the expression seen in panel E be quantified using image analysis software to confirm?

Figure 6: Do the N numbers in 6d represent the number of different individuals or replicates from the same patient?

Reviewer #2 (Remarks to the Author):

This paper describes regulation of macrophage glycolysis by fungal melanin during infection of human macrophages by *Aspergillus fumigatus*. This is an important fungal disease and the suggestion that melanin may regulate glycolysis is important in understanding the pathogenicity of this disease.

Although the paper is generally of high enough interest and the experimental work is of a very high standard suitable for publication in Nature Communications there are some issues with the paper in its current form that stand against its publication. These are outlined below.

One problem is that there is little discussion of the recent Nature Microbiology/Cell Host Microbe papers, involving several of the same authors, also describing the effect of melanin on macrophage metabolism and phagosomal development. The authors need to discuss this publication in the context of the results presented here eg how do their results fit in with reported effects of fungal melanin on NADPH oxidase or LC3 phagocytosis etc and suggest any possible overlaps. It is of course difficult to coordinate the contents of publications when they are written in parallel without knowing which paper will reach publication first – but since the previous papers are now published (and referenced here) it is hard to avoid considering them when reading the current manuscript. In fact the previous paper reports significant effects of melanin on calcium signalling and development of phagosomes via interaction with calmodulin – this is mentioned but should be discussed in detail. Sometimes the same inhibitors are used - eg W7. Were the effects

consistent with previous findings? Does fungal melanin really do all of these things in a separate manner or is it really a unified mechanism?

This issue counts against the paper as it limits the implications of the findings which may make it less relevant to the wider community.

Regulation of glycolysis or other energy metabolism by CaM is extremely well studied and therefore the changes in glycolysis observed are rather unsurprising. Also given that the previous paper shows that melanin modulates LC3 phagosomal development and hence presumably fungal survival, metabolic processes and development (eg spore germination, swelling etc) relevant to the current manuscript it would be useful to discuss these results in that context

The authors need to confirm that all isolates and strains of *A. fumigatus* used are isogenic or from the same parental strain – different isolate lineages of the fungus are known to elicit dramatically different effects from co-cultured immune cells.

It is concerning that RNAseq levels in the figures are reported incorrectly as fluorescence intensity...this needs to be corrected to reflect read count or similar.

For the genetic analysis to support a role for glycolysis several experiments are presented in Figure 6. This shows the relevance of SNPs in *PFKFB3* in disease. It is not clear from the data that the SNPs in this gene are genetically significantly associated with aspergillosis (see comment below) – or at least this data is not explicitly shown here. This argues against an important role for this gene in disease. The gene was discovered by QTL analysis and may certainly play some role, however lack of any strong statistical genetic association with aspergillosis is troubling. Was the interrogated QTL dataset created from the same cell type and challenge? Surely analysis of the CaM variant or other variants in the TOR/Akt signalling axis might have been more relevant? Statistical methods are generally not well described in the methods section itself and this needs to be addressed.

Fig 6 A and B – it is not clear what the p values refer to? – the data points are highly varied and this is for a log₂ plot – presumably the untransformed data points are even more widely spread?

Fig 6 C – the differences are mostly not significant – this should be moved to supplementary. There is marginal significance for the TT allele in donors but n=2 and 0.02 is marginal for genetic association. The data should be subjected to Cox's regression to account for age effects in the study population – it is not clear from the text that this was done?. There is some effect on HR as shown in Supp Table 3 and expression of the gene changes during infection with albino aspergillus compared with wild type but this is not enough to justify the authors strong statements that:

“We now demonstrate that SNPs in *PFKFB3* also contribute to IPA via molecular mechanisms influencing the metabolic homeostasis of immune cells.”

and

“The contribution of glucose homeostasis to antifungal immunity is supported by genetic variants in the glycolytic enzyme *PFKFB3* that act as cytokine quantitative trait loci and predispose to invasive pulmonary aspergillosis (IPA).”

Therefore the authors need to soften their statements regarding the importance of this gene in this

interaction. Selection of this gene needs to be more carefully justified in the text.

The suggestion that mTOR plays a role in transducing the melanin/calcium signal is credible and supported by several lines of evidence. This is, of itself, an important finding, but the TOR Akt signalling axis controls many aspects of cellular function not just glycolysis. For example it is also possible to state the hypothesis that TOR /Akt signalling has its primary effect on cell cycle progression or apoptosis and this affects the immune response somehow and that the change in glycolysis is a secondary consequence.

Conversely, interpretation of results involving inhibition of glycolytic function by 2 DG does not take into account possible effects due to inhibition of fungal hexokinase and the likely strong impact on fungal survival, secondary metabolite and mycotoxin production –

2DG may also have a wide ranging effect on metabolism depending on the medium. In fact 2DG does not precisely change glycolysis but inhibits the first step by blocking hexokinase – this would also affect the pentose phosphate pathway via G6P availability. If the cell then relies on glutamine as C source (or other amino acids from RPMI or DMEM) then several other areas of metabolism – gluconeogenesis, glyoxylate bypass, pentose phosphate, several areas of AA metabolism etc will be regulated to compensate – the impact of these other metabolic pathways is not considered although any or all of them could be crucial to the infection process. Different C sources such as lipid surfactant would be available in lungs. More precision in interpreting the results of these experiments and discussion of the effects is required in order to specify an effect on glycolysis alone. The use of lactate level as a marker for glycolysis and measurement of the metabolite levels is useful but does not rule out the involvement of other areas of metabolism. This is OK if the authors wish to simply say that glycolysis is affected without reference to the rest of cellular metabolism but does not tie glycolysis to the other observed effects such as cytokine production etc in a causal manner.

Therefore the authors need to address the issue and confirm that glycolysis is the primary target of the TOR signal and that glycolysis specifically is responsible for changes in cytokine secretion and the other observed effects.

Specific questions would be:

Why was the focus on glycolysis? It is certainly central to metabolism but is clearly not the only pathway that might be affected. Does the mTOR inhibition just affect glycolysis or are other pathways/processes affected.

Regulation of this pathway would generally be thought to occur by alteration of the ADP/ATP balance or by change in the environmental glucose level. Were these measured?

Are the metabolite levels in Fig 1G extracellular or intracellular, human or fungal?

It is surprising that macrophages retain 100% viability 24h after inoculation with live fungus – this is probably due to the low inoculum used (?) but some further comment is needed – but if all extracellular fungi have been eliminated it would preclude a role for fungal glucose use – fungi typically exhaust a 2% glucose medium in 24-36h - in triggering glycolysis in the human cells.

I think that some or most of these issues could be addressed by simply rewriting the relevant sections with a clearer focus on the strengths and limitations of the approach chosen.

Overall clarification of the evidence and causative steps in the signalling process described, with

more focus on the limits of the experiments performed and possible areas of uncertainty would significantly improve the paper. In my opinion better defining the limits of how well their hypothesis is supported and highlighting areas that are not well supported would still result in a paper of great interest that significantly advances knowledge in this area.

Reviewer #3 (Remarks to the Author):

Reviewers' comments:

The study reports a role of fungal melanin in activation of macrophages. The paper elucidates the dramatic effect of melanin deletion on the responses of macrophages to induce glycolysis largely in vitro and to some extent in-vivo in an infection model. The authors demonstrated a molecular link between calcium sequestration by melanin inside the phagosome and induction of glycolysis and macrophage. These findings hold a lot of promise and add to growing area of immunometabolism and calcium signaling in innate immune cells. The study involves the use of mutant strains of *A. fumigatus* for melanin or the melanin biosynthetic pathway, and purified fungal melanin, to conclusively demonstrate the role of melanin in macrophage immune response. Overall, the study is systematically performed and the evidence is extension of the key earlier findings of this group in the area of cellular metabolism and antifungal immune responses. My concerns involve some overall issues with the system and some of the interpretation can be strengthened with additional experiments. Additionally, the experiments studying calcium signaling are very confusing

1. The authors have used the total number of CD45+ cells and macrophages in the lungs of infected mice (Figure S1B) to support their claim that *A. fumigatus* infection did not display a significant loss of viability. This may be misleading, as the increased cell numbers evident in figure S1 is most likely due to lung inflammation. Author should perform flow based assays to detect cell death markers. Additionally, comparison with 2-DG treatment will substantiate authors' claim.

2. What will happen if infiltrating macrophage and alveolar macrophages are used instead of spleen macrophages (Fig. 2)? Will they show similar effect on conidiacidal activity as shown using splenocytes? There is ample evidence that local microenvironment associated drives primary and secondary pneumonia associated immune dysregulation.

3. Related to this, it would be important to recapitulate the results for glycolytic pathway and inflammatory mediators in more relevant cells i.e. alveolar and infiltrating macrophages. This is important because analyzing the whole lung homogenate, although provides an overall general effect, does not explain the macrophage specific events, which is the main focus of this study.

4. In earlier studies the authors have demonstrated that Ca²⁺ sequestration by *Aspergillus* melanin inside the phagosome abrogates activation of Ca²⁺-CaM signaling to inhibit LC3-associated phagocytosis (LAP), a non-canonical autophagy pathway (Kyrnizi I, Nat Microbiol. 2018). The author should explain the significance of this finding in the context of the current finding as LAP integrates autophagy and metabolism in the context of innate immune cell function.

5. The experiments studying calcium signaling (Figure 5) are confusing.

A) It was surprising to see the observed sustained calcium signaling for more than 1 hr. Calcium entry is generally transient and usually cells return to baseline within 5-10 min. Any explanation for this rather unusual phenomenon will be helpful. Perhaps a ratio metric dye should be used.

B) The use of EGTA is OK, but unless the cells are stimulated there will be no calcium entry for lactate release. It thus does not preclude calcium entry. The experiments should have been performed with and without Thapsigargin. Also, the authors' need to differentiate store depletion versus calcium entry, by using 0 calcium outside followed by addition of 2 mM calcium to record an ER release and corresponding calcium entry in these cells.

C) Although it is true that ER calcium modulates phagosomes calcium entry, this needs to be further evaluated using Fura2 conjugated zymosan particles. Alternately could be mediated by STIM1 rather than calmodulin per se, and in that case STIM1 should be silenced to see its effect.

Minor:

6. I had trouble reconciling some of the data where the indicated changes, contrary to authors' claim, are rather modest (Fig. 5E).

Reviewer #4 (Remarks to the Author):

This manuscript describes the metabolic changes occurring in macrophages during infection with *Aspergillus fumigatus*. The authors propose that fungal melanin promotes glycolysis and anti-fungal responses by remodeling the intracellular calcium machinery and impairing signaling via calmodulin. They also propose that calcium/CaM signaling orchestrates mTOR recruitment to the phagosome during infection, and that this effect is directly regulated by fungal melanin.

The manuscript is interesting and, at least in some parts, convincing. However, a few statements are not strongly supported by the data and there are major problems in a few key figures.

In details:

In most of the figures, control values are missing. For example: which is the value of lactate secretion in unstimulated cells in each experiment? And mRNA in unstimulated cells? Every experiment must have its own control. For example: how do we know that in Fig S3D the cells are responding? What is the effect of the drugs used in Fig 5 on control cells (cells stimulated with $\Delta ku80$ strain)?

The fact that in most of the experiments proper controls are missing makes the entire manuscript weak.

Figure 3F: the authors performed a few experiments to exclude the involvement of the melanin receptor MelLec in the process. They stimulated macrophages with chemically synthesized 1,8-DHN and "observed that the levels of lactate secreted by macrophages stimulated with 1,8-DHN were significantly lower than those obtained after infection with the $\Delta ku80$ strain (Figure 3F), suggesting that recognition of 1,8-DHN melanin and signalling via MelLec is, to a large extent, dispensable for the activation of glycolysis." However, this experiment is not convincing: it would be necessary to perform a dose-response curve. Moreover, the results would be more convincing in the presence of data showing MelLec triggering in the experimental conditions used by the authors.

Figure 4 and HIF-1 α : Fig 4E shows immunofluorescence in very few cells (from 1 to 3 per condition) and the results must to be quantified and expressed as HIF staining in nuclei/total HIF staining. Also, Fig S4E doesn't show significant inhibition of HIF-1 α accumulation in lysates. Considering these data, the involvement of HIF-1 α is questionable. Fig 4F is potentially very strong but proper controls are missing: it is important to show that HIFc macrophages recognize $\Delta ku80$ (phagocytosis). Moreover, is the expression of MelLec equivalent?

Figure 5: this figure is crucial for the manuscript but – again – lacks proper controls.

What about EGTA, TG, W7 in cells stimulated by $\Delta ku80$?

Does calcium manipulation affect recognition of pathogens? And phagocytosis? Probably yes, so how to interpret the results shown in this figure?

EGTA 500 μ M to buffer extracellular calcium in DMEM is inadequate. Unless calcium-free medium is used, the experiments should be performed using 5mM EGTA.

Was the pH controlled?

Which statistical analysis was used in each figure 5 and why? (especially 5D, F, G).

In Fig 5F the immunofluorescence doesn't make any point and the arrows in the first panel do not show mTOR accumulation. Concerning the image analysis, it is stated: "Data on quantification of mTOR+ phagosomes was determined by analyzing at least 200 phagosomes." But how was the analysis performed?

Point-by-point reply letter

Reviewer #1

Reviewer's comment: *This study seeks to make a link between the fungal cell wall and host cell metabolism, two major topics in fungal host-pathogen interactions. The authors show a link between fungal melanin, host calcium flux, mTOR and HIF1alpha and Warburg shift. I thought the introduction was excellent.*

Author's reply: We thank the reviewer for his/her positive comments on our manuscript.

Reviewer's comment: *Figure 1: How many human macrophage replicates were undertaken from how many individuals?*

Author's reply: Unless stated otherwise, we have used three technical replicates from each individual in the experiments involving human macrophages. Therefore, in the graphs referring to data on human macrophages, each dot represents the mean value of three technical replicates obtained for each individual (i.e., in the figure legends the n refers to the number of individuals). In the transcriptomics and metabolomics analyses, pooled technical replicates from three different individuals were analyzed in each experimental condition. We have clarified this issue in the Methods section, under *Cell stimulations and treatments, RNA sequencing and LC-MS/MS targeted metabolomics*.

Reviewer's comment: *Were the macrophages infected with resting or swollen conidia or hyphae and at what MOI? Figure 1H the loss of viability suggests a low MOI and resting conidia. This is corroborated by the down regulation of cytokines at 6 hours. Are you sure these macrophages are actually infected at later time points?*

Author's reply: Macrophages were infected with resting conidia, unless stated otherwise (e.g., in Fig. 3a). The MOI used was 1:10 (macrophage: conidia), with the exception of the data presented in Fig.1f that also included an experimental condition using a MOI of 1:2, and the protocols involving the determination of the fungicidal activity and phagocytosis, which are performed using different conditions described in detail in the Methods section. As for the standard infection model, we have confirmed that the infection of macrophages is sustained throughout the experiment (Fig.1 of this reply letter). Of note, and despite fungal germination is observed at later time points, only minor macrophage death is observed, as revealed by propidium iodide staining.

Fig. 1- Human macrophages infected with FITC-labelled resting conidia of *A. fumigatus* (green) at a MOI of 1:10 for 4, 8 and 24 hours. After infection, macrophages were stained with DAPI (blue) and propidium iodide (red).

Reviewer's comment: Likewise, I assume 1×10^8 inoculum was used for the animal experiments but how pathogenic is the *Ku* strain and why wasn't CEA10 used?

Author's reply: We confirm that an inoculum of 1×10^8 live conidia was used, as stated in the Methods section, under *Mouse infection*. We have used the A1163 $\Delta ku80$ strain in order to maintain the consistency across experiments throughout the paper, since the remaining deletion mutants used throughout this study (e.g., $\Delta pksP$ and $\Delta pksP/rodA$) were generated in this background [with the exception of the mutant strains in the DHN-melanin biosynthetic pathway, which were generated in the B-5233 background (Supplementary Fig. 3b)]. The deletion of the *akuB* (*ku80*) gene has been widely reported to have no significant effect on virulence (for example, da Silva Ferreira ME et al. Eukaryot Cell. 2006). However, given the reviewer's concern, we have compared the virulence profile of the CEA10 and $\Delta ku80$ strains in an immunosuppressed mouse model. Our results show no significant impact of the *akuB* deletion on fungal pathogenicity (Fig. 2 of this reply letter).

Fig. 2- Survival (%) of cyclophosphamide-treated mice (200 mg/kg) at day -4 and day-1 and either left untreated (Ctrl) or infected intranasally with 5×10^5 live conidia of the CEA10 or $\Delta ku80$ strains of *A. fumigatus* (n=8 in each group).

Reviewer's comment: *Figure 2: 2B and D seem to show that all the phagocytosed conidia are killed, presumably the non phagocytosed conidia survive. So, the issue seems that it is not clear what level of productive and sustained macrophage infection there is in the model.*

Author's reply: As described in the Methods section, under *Phagocytosis assay* and *Conidiacidal activity assay*, the MOI used for the killing and phagocytosis assays were different (MOI of 10:1 and 1:5, respectively). It is thus not possible to infer from Fig. 2b and 2d that all the phagocytosed conidia were killed. Indeed, as shown in Fig. 1 of this reply letter, whereas all conidia were internalized by macrophages using a MOI of 1:10 (the same used for cytokine and lactate measurements), a proportion of them was however able to survive, as revealed by their ability to germinate. Taken together, these observations support a model based on sustained infection.

Reviewer's comment: *How specific is 2DG for glycolysis?*

Author's reply: 2-DG is a stable, non-metabolizable glucose analog that inhibits phosphorylation of glucose by hexokinase, mimicking glucose deprivation conditions. The blockade of the hexokinase suggests therefore that 2-DG could also affect the pentose phosphate pathway (PPP). In line with the reviewer's concern, and to exclude any potential side effects in our results, we have also tested the use of galactose, that reduces net ATP yield from glycolysis without blocking its flux, and medium without glucose (see Supplementary Fig. 2b-d), with similar results to those obtained using 2-DG. Furthermore, we have now complemented these analyses with the use of 3PO, which by inhibiting PFKFB3 and decreasing the levels of fructose-2,6-bisphosphate, selectively blocks glycolysis, and 6-AN, an inhibitor of the 6-phosphogluconate dehydrogenase enzyme of the PPP pathway. Our results show a significant decrease in lactate secretion (new Fig. 2f of the revised manuscript) and cytokine production (new Supplementary Fig. 2f of the revised manuscript) by human macrophages treated with 3PO, but not 6-AN, confirming therefore the dominant role of glycolysis in our model.

Reviewer's comment: *Figure 3: The issue of resting vs swollen conidia is now addressed. This reveals that melanin induces glycolysis and lactate production. The genetics is convincing.*

Author's reply: We thank the reviewer for his/her positive comments on our manuscript.

Reviewer's comment: *Figure 4: How many replicates were used for the transcriptomics from how many patients? Can the expression seen in panel E be quantified using image analysis software to confirm?*

Author's reply: In the transcriptomics analyses, pooled technical replicates from three different individuals in each experimental condition were analyzed. We have clarified this in the Methods section, under *RNA sequencing*. In addition, and following the reviewer's suggestion, we have now confirmed the expression of HIF-1a using the image analysis software ImageJ (new Supplementary Fig. 4e of the revised manuscript). Additional information has also been added to the Methods section, under *Live cell imaging*.

Reviewer's comment: *Figure 6: Do the N numbers in 6d represent the number of different individuals or replicates from the same patient?*

Author's reply: Following the editor's recommendation, we have now removed entirely the section describing the genetic analyses implicating genetic variation in PFKFB3 in the metabolic reprogramming of macrophages and susceptibility to invasive pulmonary aspergillosis.

Reviewer #2

Reviewer's comment: *This paper describes regulation of macrophage glycolysis by fungal melanin during infection of human macrophages by *Aspergillus fumigatus*. This is an important fungal disease and the suggestion that melanin may regulate glycolysis is important in understanding the pathogenicity of this disease. Although the paper is generally of high enough interest and the experimental work is of a very high standard suitable for publication in *Nature Communications* there are some issues with the paper in its current form that stand against its publication. These are outlined below.*

Author's reply: We thank the reviewer for his/her positive comments on our manuscript.

Reviewer's comment: *One problem is that there is little discussion of the recent *Nature Microbiology/Cell Host Microbe* papers, involving several of the same authors, also describing the effect of melanin on macrophage metabolism and phagosomal development. The authors need to discuss this publication in the context of the results presented here e.g. how do their results fit in with reported effects of fungal melanin on NADPH oxidase or LC3 phagocytosis etc and suggest any possible overlaps. It is of course difficult to coordinate the contents of publications when they are written in parallel without knowing which*

paper will reach publication first – but since the previous papers are now published (and referenced here) it is hard to avoid considering them when reading the current manuscript. In fact, the previous paper reports significant effects of melanin on calcium signalling and development of phagosomes via interaction with calmodulin – this is mentioned but should be discussed in detail.

Author's reply: We agree with the reviewer that the overlap between melanin-regulated mechanisms, particularly those implicating calcium signalling and coordinating the activation of LC3-associated phagocytosis (LAP) and cellular metabolism could be discussed in more detail. In the context of our previous publications mentioned by the reviewer, we have now addressed the role of melanin in the activation of LAP and the metabolic reprogramming of macrophages in the Discussion section (page 14, second paragraph).

Reviewer's comment: Sometimes the same inhibitors are used – e.g. W7. Were the effects consistent with previous findings?

Author's reply: The effects of the W7 inhibitor in preventing CaM-mediated signaling and the activation of LAP were recapitulated in our model of macrophage infection. Using this model, the use of W7 promoted a decrease in the activation of LAP, as revealed by the lower percentage of LC3+ phagosomes in macrophages after infection with the $\Delta pksP$ strain (Fig. 3 of this reply letter). These results are in accordance to what we previously demonstrated in monocytes (Kyrnizi et al, Nat Microbiol 2018), confirming that the blockade of CaM-mediated signalling impairs the activation of LAP by albino conidia, while instead, it rescues the inability of the $\Delta pksP$ strain to induce glycolysis in infected macrophages.

Figure 3. LC3-associated phagocytosis (LAP) in primary human macrophages upon infection with live resting conidia of $\Delta ku80$ or $\Delta pksP$ strains of *A. fumigatus* for 3 hours (MOI of 1:3), with or without the presence of the CaM antagonist W7 (25 μ M) added 10 min post-infection. The percentage of LC3+ phagosomes was determined by confocal microscopy and the results are presented as the mean \pm SEM of three independent experiments.

Reviewer's comment: Does fungal melanin really do all of these things in a separate manner or is it really a unified mechanism? This issue counts against the paper as it limits the implications of the findings which may make it less relevant to the wider

community.

Author's reply: As stated above and considering our previously published data as well as the results presented in this manuscript, fungal melanin represents a central molecule that regulates both the activation of LAP and the metabolic reprogramming of macrophages via shared molecular mechanisms (e.g., calcium sequestration). Therefore, fungal melanin appears to represent a “unified mechanism” with (at least) two important and distinct consequences for the outcome of the host-fungus interaction, as described above and in the Discussion section. The finding that purified melanin from different sources with heterogeneous chemical and physical properties, and not only fungal melanin, recapitulates the inhibition of LAP (Akoumaniaki et al, Cell Host Microbe, 2016) and the induction of glycolysis (data not shown) further supports the relevance of these findings to the general scientific community. The dual role of melanin in regulating both LAP and the metabolic reprogramming of macrophages was addressed in more detail in the Discussion section (page 14, second paragraph).

Reviewer's comment: *Regulation of glycolysis or other energy metabolism by CaM is extremely well studied and therefore the changes in glycolysis observed are rather unsurprising. Also given that the previous paper shows that melanin modulates LC3 phagosomal development and hence presumably fungal survival, metabolic processes and development (eg spore germination, swelling etc) relevant to the current manuscript it would be useful to discuss these results in that context.*

Author's reply: We share the reviewer's opinion that fungal germination and swelling are essential processes for the reprogramming of macrophage metabolism. Our hypothesis is that, by blocking the activation of LAP, melanin contributes to fungal persistence within the phagosome, allowing conidial germination and active removal of melanin that, in line with our results, provides the necessary signals to reorient macrophage metabolism towards glycolysis. This concept was further highlighted in the Discussion section (page 14, second paragraph).

Reviewer's comment: *The authors need to confirm that all isolates and strains of A. fumigatus used are isogenic or from the same parental strain – different isolate lineages of the fungus are known to elicit dramatically different effects from co-cultured immune cells.*

Author's reply: We agree with the reviewer that different fungal isolates could elicit different immune responses. We confirm that the *A. fumigatus* strains used in this study are derived from the same parental strain. Specifically, the cell wall mutants $\Delta pksP$ and $\Delta rodA/pksP$ used throughout the manuscript were generated using $\Delta ku80$ as the parental strain, whereas the mutants in the DHN-melanin biosynthetic pathway were generated in the B-5233 background. In these conditions, the $\Delta ku80$ or the B-5233 strains

were used as wild-type controls, respectively. This information is available in the Methods section, under *Aspergillus strains and culture conditions*.

Reviewer's comment: *It is concerning that RNAseq levels in the figures are reported incorrectly as fluorescence intensity...this needs to be corrected to reflect read count or similar.*

Author's reply: We apologize if the results presented in the figures were not entirely clear. However, and in contrast to the reviewer's statement, we were not able to identify which figures related to the RNAseq data were reported as fluorescence intensity. More specifically, gene expression levels determined by RNAseq were reported as expression (log2 fold) vs adjusted P value in Fig. 1a or as normalized relative expression (log2 fold) and presented as heatmaps in Figs. 1c and 4a.

Reviewer's comment: *For the genetic analysis to support a role for glycolysis several experiments are presented in Figure 6. This shows the relevance of SNPs in PFKFB3 in disease. It is not clear from the data that the SNPs in this gene are genetically significantly associated with aspergillosis (see comment below) – or at least this data is not explicitly shown here. This argues against an important role for this gene in disease. The gene was discovered by QTL analysis and may certainly play some role, however lack of any strong statistical genetic association with aspergillosis is troubling. Was the interrogated QTL dataset created from the same cell type and challenge? Surely analysis of the CaM variant or other variants in the TOR/Akt signalling axis might have been more relevant? Statistical methods are generally not well described in the methods section itself and this needs to be addressed.*

Author's reply: Following the editor's recommendation, we have now removed entirely the section described the genetic analyses implicating genetic variation in PFKFB3 in the metabolic reprogramming of macrophages and susceptibility to invasive pulmonary aspergillosis.

Reviewer's comment: *Fig 6 A and B – it is not clear what the p values refer to? – the data points are highly varied and this is for a log2 plot – presumably the untransformed data points are even more widely spread?*

Author's reply: Following the editor's recommendation, we have now removed entirely the section described the genetic analyses implicating genetic variation in PFKFB3 in the metabolic reprogramming of macrophages and susceptibility to invasive pulmonary aspergillosis.

Reviewer's comment: *Fig 6 C – the differences are mostly not significant – this should be moved to supplementary. There is marginal significance for the TT allele in donors but n=2 and 0.02 is marginal for genetic association. The data should be subjected to Cox's regression to account for age effects in the study population – it is not clear from the text that this was done?. There is some effect on HR as shown in Supp Table 3 and expression of the gene changes during infection with albino aspergillus compared with wild type but this is not enough to justify the authors strong statements that: "We now demonstrate that SNPs in PFKFB3 also contribute to IPA via molecular mechanisms influencing the metabolic homeostasis of immune cells. and "The contribution of glucose homeostasis to antifungal immunity is supported by genetic variants in the glycolytic enzyme PFKFB3 that act as cytokine quantitative trait loci and predispose to invasive pulmonary aspergillosis (IPA). "Therefore the authors need to soften their statements regarding the importance of this gene in this interaction. Selection of this gene needs to be more carefully justified in the text.*

Author's reply: Following the editor's recommendation, we have now removed entirely the section described the genetic analyses implicating genetic variation in PFKFB3 in the metabolic reprogramming of macrophages and susceptibility to invasive pulmonary aspergillosis.

Reviewer's comment: *The suggestion that mTOR plays a role in transducing the melanin/calcium signal is credible and supported by several lines of evidence. This is, of itself, an important finding, but the TOR /Akt signalling axis controls many aspects of cellular function not just glycolysis. For example it is also possible to state the hypothesis that TOR /Akt signalling has its primary effect on cell cycle progression or apoptosis and this affects the immune response somehow and that the change in glycolysis is a secondary consequence.*

Author's reply: We agree with the reviewer regarding the possibility that mTOR/Akt signaling could have its primary effect on processes such as cell cycle progression or apoptosis, affecting the immune response. In line with the reviewer's concern, we have now clarified this point. By resorting to human macrophages left uninfected or infected with the $\Delta ku80$ strain in the presence of mTOR (rapamycin) or Akt (wortmannin) inhibitors, we demonstrate that neither cell cycle progression nor cellular viability were significantly affected by the blockade of these pathways (Fig. 4 of this reply letter).

Figure 4. Cell cycle progression (left) and viability (right) of human macrophages infected with live conidia of the $\Delta ku80$ strain of *A. fumigatus* for 6 hours (MOI of 1:5) with or without mTOR (rapamycin, 10 nM) or Akt (wortmannin, 10 μ M), added 1 hour prior to infection. Results were analyzed by flow cytometry and expressed as percentage of cells (mean of two independent experiments).

Reviewer's comment: Conversely, interpretation of results involving inhibition of glycolytic function by 2-DG does not take into account possible effects due to inhibition of fungal hexokinase and the likely strong impact on fungal survival, secondary metabolite and mycotoxin production.

Author's reply: We share the reviewer's opinion that this would be an important point to consider. However, it is noteworthy that 2-DG was added as a pre-treatment and removed afterwards from the medium prior to the infection. Therefore, it is unlikely that 2-DG could act on fungal metabolism with an impact on survival. Further supporting this hypothesis, the number of fungal CFUs recovered after infection of macrophages increases in a 2-DG dose-dependent manner (Fig. 2b). Nevertheless, and given the reviewer's concern, we further tested the direct effect of 2-DG on the survival of fungal spores (Fig. 5 of this reply letter). As revealed by the use of the FUN1 viability stain, no significant consequences of the 2-DG treatment on fungal survival were observed. In the same conditions, the expression of fungal

hexokinase *hxkA* was also not significantly altered (Fig. 6 of this reply letter).

Figure 5. Fungal viability after treatment with 2-DG. Representative immunofluorescence images of live conidia of the $\Delta ku80$ strain of *A. fumigatus* stained with FUN1 and cultured in RPMI for 4 and 8 hours with or without the presence of 2-DG (10 mM). DMSO was used as positive control. Live conidia are indicated in red and dead conidia are indicated in green.

Figure 6. Expression of *hxkA* after treatment with 2-DG. Live conidia of the $\Delta ku80$ strain of *A. fumigatus* were cultured in complete RPMI for 1 hour with or without the presence of 2-DG (10 mM) and expression of *hxkA* was analyzed by qRT-PCR. Data are presented as mean \pm SEM of two independent experiments performed in triplicate.

Reviewer's comment: *2DG may also have a wide ranging effect on metabolism depending on the medium. In fact, 2DG does not precisely change glycolysis but inhibits the first step by blocking hexokinase – this would also affect the pentose phosphate pathway via G6P availability. If the cell then relies on glutamine as C source (or other amino acids from RPMI or DMEM) then several other areas of metabolism – gluconeogenesis, glyoxylate bypass, pentose phosphate, several areas of AA metabolism etc will be regulated to compensate – the impact of these other metabolic pathways is not considered although any or all of them could be crucial to the infection process. Different C sources such as lipid surfactant would be available in lungs. More precision in interpreting the results of these experiments and discussion of the effects is required in order to specify an effect on glycolysis alone. The use of lactate level as a marker for glycolysis and measurement of the metabolite levels is useful but does not rule out the involvement of other areas of metabolism. This is OK if the authors wish to simply say that glycolysis is affected without reference to the rest of cellular metabolism but does not tie glycolysis to the other observed effects such as cytokine production etc in a causal manner. Therefore, the authors need to address the issue and confirm that glycolysis is the primary target of the TOR signal and that glycolysis specifically is responsible for changes in cytokine secretion and the other observed effects.*

Author's reply: We understand the reviewer's concern that 2-DG could have a wide range of effects on cellular metabolism, by not exclusively influencing glycolysis but also the PPP via G6P availability. To further clarify the role of glycolysis in our system, we have now complemented our analyses with the use of 3PO, which by inhibiting PFKFB3 and decreasing the levels of fructose-2,6-bisphosphate, selectively blocks glycolysis, and 6-AN, an inhibitor of the 6-phosphogluconate dehydrogenase enzyme of the PPP. Our results show a significant decrease in lactate secretion (new Fig. 2f of the revised manuscript) and cytokine production (new Supplementary Fig. 2f of the revised manuscript) by human macrophages treated with 3PO, but not 6-AN, to an extent comparable to that of 2-DG. These results complement the findings already presented in the manuscript using galactose, that reduces net glycolysis ATP yield without blocking glycolysis flux, and medium without glucose (see Supplementary Figs. 2b-d) and that highlighted glycolysis as the dominant metabolic pathway in our model. However, and as referred by the reviewer, we cannot entirely rule out the relevant contribution of metabolic pathways other than glycolysis or the PPP in antifungal immunity and this has been pointed out in the Discussion (page 14, third paragraph).

Reviewer's comment: *Specific questions would be: Why was the focus on glycolysis? It is certainly central to metabolism but is clearly not the only pathway that might be affected. Does the mTOR inhibition just affect glycolysis or are other pathways/processes affected.*

Author's reply: We focused our efforts on the clarification of the role of glycolysis in antifungal immunity since it has been vastly described in the literature as a major metabolic pathway involved in the activation of macrophage responses against pathogens. However, and as referred by the reviewer, this does not preclude that other metabolic pathways (some of them currently under investigation in the laboratory), and likely interacting with glycolysis, are also relevant for antifungal immunity. For this reason, we have now rephrased the Discussion in order to account for the potential importance of metabolic pathways other than glycolysis (page 14, third paragraph). In addition, we agree that mTOR inhibition could have a likely impact on a broad range of pathways and processes. However, our assumption that glycolysis is relevant for antifungal immunity is not derived only from the experiments that targeted mTOR inhibition but involves additional extensive in vitro and in vivo data to support its involvement. Nevertheless, and although other processes may still be affected, mTOR inhibition does not impact cell cycle progression or cellular viability, as demonstrated above (Fig. 4 of this reply letter).

Reviewer's comment: *Regulation of this pathway would generally be thought to occur by alteration of the ADP/ATP balance or by change in the environmental glucose level. Were these measured?*

Author's reply: We agree that the ADP/ATP balance could provide critical insights about the regulation of glycolysis. We have now measured this parameter in infected macrophages and demonstrate that fungal infection elicits an increase in the ADP/ATP ratio (new Fig. 1g of the revised manuscript), a finding consistent with the activation of glycolysis. Along the same line, infected macrophages also displayed an enhanced consumption of glucose (data already available in Fig. 1f of the manuscript).

Reviewer's comment: *Are the metabolite levels in Fig 1G extracellular or intracellular, human or fungal?*

Author's reply: The metabolite levels in Fig. 1g (now Fig.1h of the revised manuscript) are intracellular, and from human origin. In our infection model, the protocol for metabolite extraction is not sufficiently harsh to cause the rupture of the rigid and highly hydrophobic fungal cell wall. Furthermore, as the relative size of the conidia compared to the macrophage is approximately 1/10, the moment that cell debris are centrifuged and removed, conidia should be removed as well. Therefore, and although this has not been specifically addressed, the contribution of the fungus to the overall metabolite pool should be negligible.

Reviewer's comment: *It is surprising that macrophages retain 100% viability 24h after inoculation with live fungus – this is probably due to the low inoculum used (?) but some further comment is needed – but if all extracellular fungi have been eliminated it would preclude a role for fungal glucose use – fungi typically exhaust a 2% glucose medium in 24-36h - in triggering glycolysis in the human cells.*

Author's reply: The data showing viability in Fig. 1h (now Supplementary Fig. 1b of the revised manuscript) is presented as relative fluorescence units and not as percentage of viability. We can thus only conclude that there are no significant differences between infected and uninfected macrophages regarding their viability. Given the reviewer's concern, we have further characterized macrophage viability using annexin/PI, and we have confirmed the lack of a significant loss of viability in infected macrophages, as compared to uninfected controls (new Supplementary Fig. 1b of the revised manuscript).

Reviewer's comment: *I think that some or most of these issues could be addressed by simply rewriting the relevant sections with a clearer focus on the strengths and limitations of the approach chosen. Overall clarification of the evidence and causative steps in the signalling process described, with more focus on the limits of the experiments performed and possible areas of uncertainty would significantly improve the paper. In my opinion better defining the limits of how well their hypothesis is supported and highlighting areas that are not well supported would still result in a paper of great interest that significantly advances knowledge in this area.*

Author's reply: We thank the reviewer for his/her valuable suggestions, and we have now modified the relevant sections accordingly (see our comments above).

Reviewer #3

Reviewer's comment: *The study reports a role of fungal melanin in activation of macrophages. The paper elucidates the dramatic effect of melanin deletion on the responses of macrophages to induce glycolysis largely in vitro and to some extent in-vivo in an infection model. The authors demonstrated a molecular link between calcium sequestration by melanin inside the phagosome and induction of glycolysis and macrophage. These findings hold a lot of promise and add to growing area of immunometabolism and calcium signaling in innate immune cells. The study involves the use of mutant strains of A. fumigatus for melanin or the melanin biosynthetic pathway, and purified fungal melanin, to conclusively demonstrate the role of melanin in macrophage immune response. Overall, the study is systematically performed and the*

evidence is extension of the key earlier findings of this group in the area of cellular metabolism and antifungal immune responses.

Author's reply: We thank the reviewer for his/her positive comments on our manuscript.

Reviewer's comment: *My concerns involve some overall issues with the system and some of the interpretation can be strengthened with additional experiments. Additionally, the experiments studying calcium signaling are very confusing.*

Author's reply: We have addressed the reviewer's concerns (see below) and we now hope that the results are presented more clearly.

Reviewer's comment: *1. The authors have used the total number of CD45+ cells and macrophages in the lungs of infected mice (Figure S1B) to support their claim that A. fumigatus infection did not display a significant loss of viability. This may be misleading, as the increased cell numbers evident in figure S1 is most likely due to lung inflammation. Author should perform flow based assays to detect cell death markers. Additionally, comparison with 2-DG treatment will substantiate authors' claim.*

Author's reply: We understand the reviewer's concern that the results presented in Supplementary Fig. 1b could indeed result from the increased lung inflammation. To clarify this point, and following the reviewer's suggestion, we have performed flow cytometry-based assays to quantify cell death using the Zombie Violet dye. The results demonstrate that there is no significant loss of cell viability upon in vivo infection with *A. fumigatus* (new Supplementary Fig. 1c of the revised manuscript). The same approach was used to demonstrate that macrophage viability after in vivo treatment with 2-DG remains unaltered (new Supplementary Fig. 2i of the revised manuscript).

Reviewer's comment: *What will happen if infiltrating macrophage and alveolar macrophages are used instead of spleen macrophages (Fig. 2)? Will they show similar effect on conidiacidal activity as shown using splenocytes? There is ample evidence that local microenvironment associated drives primary and secondary pneumonia associated immune dysregulation. Related to this, it would be important to recapitulate the results for glycolytic pathway and inflammatory mediators in more relevant cells i.e. alveolar and infiltrating macrophages. This is important because analyzing the whole lung homogenate, although provides an overall general effect, does not explain the macrophage specific events, which is the main focus of this study.*

Author's reply: We agree with the reviewer that the influence of the local microenvironment could affect the antifungal immune responses and should be considered. Therefore, following the reviewer's suggestion, we have FACS-sorted both alveolar and infiltrating macrophages from naïve mice (gating strategy depicted in new Supplementary Fig. 2g). We were unable to test the functional activity of cells sorted directly from infected animals, since a proportion of the recovered cells was already found to contain several ingested conidia, and this compromised data analysis. Infection of sorted cells from naïve animals with the $\Delta ku80$ strain of *A. fumigatus* was nonetheless found to induce lactate secretion by both alveolar (new Fig. 2g of the revised manuscript) and infiltrating (Fig. 7 of this reply letter) macrophages. Importantly, the levels of lactate secreted were significantly decreased following treatment with 2-DG in both cell types, a finding consistent with the decreased levels of TNF- α (Fig. 7 of this reply letter) produced by infiltrating macrophages (we were unable to detect cytokine production by alveolar macrophages). Collectively, these results recapitulate those obtained with whole lung homogenates and splenic macrophages and discard a potential confounding effect of the cell type on the reprogramming of cellular metabolism.

Figure 7. The glycolytic pathway is required for protective immune responses to *A. fumigatus* in infiltrating macrophages. Infiltrating macrophages were FACS-sorted from mice and after a resting period of 18 hours, cells were infected with *A. fumigatus* for 24 hours (MOI of 1:10). Lactate secretion and cytokine production were assessed in culture supernatants. In some conditions, macrophages were pre-treated with 10 mM of 2-DG for 1 hour.

Reviewer's comment: *In earlier studies the authors have demonstrated that Ca²⁺ sequestration by Aspergillus melanin inside the phagosome abrogates activation of Ca²⁺-CaM signaling to inhibit LC3-associated phagocytosis (LAP), a non-canonical autophagy pathway (Kyrnizi I, Nat Microbiol. 2018). The author should explain the significance of this finding in the context of the current finding as LAP integrates autophagy and metabolism in the context of innate immune cell function.*

Author's reply: We agree with the reviewer that the overlap between melanin-regulated mechanisms coordinating the activation of LC3-associated phagocytosis (LAP) and cellular metabolism could be discussed in more detail. In the context of our previous publications mentioned by the reviewer, we have now added information to the Discussion section addressing the role of melanin in the activation of LAP and the metabolic reprogramming of macrophages. Our hypothesis is that, by blocking the activation of LAP, melanin contributes to fungal persistence within the phagosome, allowing conidial germination and active shedding of melanin that, in line with our results, provides the necessary signals to reorient

macrophage metabolism towards glycolysis. This concept was highlighted in the Discussion section (page 14, second paragraph) (see also reply to reviewer #2).

Reviewer's comment: *The experiments studying calcium signaling (Figure 5) are confusing. A) It was surprising to see the observed sustained calcium signaling for more than 1 hr. Calcium entry is generally transient and usually cells return to baseline within 5-10 min. Any explanation for this rather unusual phenomenon will be helpful. Perhaps a ratio metric dye should be used.*

Author's reply: We understand the reviewer's concern regarding the observed sustained calcium signaling. However, the pattern of calcium signaling that is observed in certain experimental conditions is dependent on the type of stimulus and may explain why we observe mostly a sustained/oscillating (and not a transient) pattern of calcium signaling (for additional examples of different patterns of calcium signaling, see Le Borge M et al, J Immunol, 2016). Nevertheless, and given the reviewer's concern, we have performed live imaging analysis of human macrophages preloaded with Fluo4-AM and infected with the $\Delta pksP$ strain. In support of a sustained pattern of calcium signalling, our results demonstrate that calcium fluxes occur up to 30-45 minutes after infection (data not shown). The same analysis reveals that periodic calcium spikes take place at least up to 15 minutes after infection (the maximum time that we have measured) with the $\Delta pksP$ strain (new Supplementary Fig. 5a of the revised manuscript). In addition, to clarify the extent of the decrease in calcium signaling after infection, we have modified the graph in Fig. 5a to include the basal results obtained for unstimulated cells. Although the results are convincing, should the reviewer still believe it would be worthwhile addressing this point with a ratiometric dye, these experiments can be performed and added to the manuscript.

B) The use of EGTA is OK, but unless the cells are stimulated there will be no calcium entry for lactate release. It thus does not preclude calcium entry. The experiments should have been performed with and without Thapsigargin. Also, the authors' need to differentiate store depletion versus calcium entry, by using 0 calcium outside followed by addition of 2 mM calcium to record an ERrelease and corresponding calcium entry in these cells.

Author's reply: In line with the reviewer's comment, the results presented in Fig. 5 were obtained using infected macrophages. However, given the reviewer's concern and also following a suggestion from reviewer #4, we have now replaced the data obtained using 500 μ M of EGTA by results using calcium-free medium alone (new Fig. 5b of the revised manuscript). Experiments performed using thapsigargin were already available (Fig. 5c). To further improve our conclusions, we have adopted the reviewer's suggestion

to differentiate store depletion versus calcium entry, and have performed infection of macrophages using calcium-free media followed by the addition of 2 mM calcium. In these conditions, the level of lactate secreted by macrophages infected with the $\Delta pksP$ strain was not rescued to an extent comparable to that obtained upon infection with the $\Delta ku80$ strain (new Fig. 5d of the revised manuscript). This result therefore demonstrates that store depletion and not calcium entry regulates intracellular signaling pathways required for the activation of glycolysis by fungal melanin.

C) Although it is true that ER calcium modulates phagosomes calcium entry, this needs to be further evaluated using Fura2 conjugated zymosan particles. Alternately could be mediated by STIM1 rather than calmodulin per se, and in that case STIM1 should be silenced to see its effect.

Author's reply: Following the reviewer's suggestion, we have tried to use Fura2-conjugated zymosan particles, as described (Lundqvist-Gustafsson et al. Cell Calcium 2000). However, we were not be able to acquire the product, at least in a reasonable timeframe to perform the experiments. Therefore, and as an alternative also suggested by the reviewer, to shed further light into the mediators involved in calcium entry into the phagosome from the ER, we have silenced STIM1 in primary human macrophages using siRNA. Following the successful silencing of *STIM1* (new Supplementary Fig. 5c of the revised manuscript), macrophages were infected with either the $\Delta ku80$ or $\Delta pksP$ strain of *A. fumigatus*. The results demonstrate that lactate secretion is not altered by the suppression of STIM1 (new Fig. 5e of the revised manuscript), further supporting the hypothesis that ER calcium stores regulate immunometabolic signaling, independently of extracellular calcium.

Reviewer's comment: *I had trouble reconciling some of the data where the indicated changes, contrary to authors' claim, are rather modest (Fig. 5E).*

Author's reply: We understand the reviewer's concern regarding the modest changes in total mTOR levels between human macrophages infected with the $\Delta ku80$ or $\Delta pksP$ strains of *A. fumigatus*. For that reason, and because our goal was actually to assess mTOR co-localization within the phagosome (Fig. 5g) and not total levels of mTOR, we have now transferred this information to the supplementary appendix (new Supplementary Fig. 5e).

Reviewer #4

Reviewer's comment: *This manuscript describes the metabolic changes occurring in macrophages during infection with *Aspergillus fumigatus*. The authors propose that fungal melanin promotes glycolysis and anti-fungal responses by remodeling the intracellular calcium machinery and impairing signaling via calmodulin. They also propose that calcium/CaM signaling orchestrates mTOR recruitment to the phagosome during infection, and that this effect is directly regulated by fungal melanin. The manuscript is interesting and, at least in some parts, convincing.*

Author's reply: We thank the reviewer for his/her positive comments on our manuscript.

Reviewer's comment: *However, a few statements are not strongly supported by the data and there are major problems in a few key figures. In details: In most of the figures, control values are missing. For example: which is the value of lactate secretion in unstimulated cells in each experiment? And mRNA in unstimulated cells? Every experiment must have its own control. For example: how do we know that in Fig S3D the cells are responding? What is the effect of the drugs used in Fig 5 on control cells (cells stimulated with $\Delta ku80$ strain)? The fact that in most of the experiments proper controls are missing makes the entire manuscript weak.*

Author's reply: We agree with the reviewer that control values are essential, and, for that reason, we have now modified the relevant figures throughout the manuscript accordingly.

Reviewer's comment: *Figure 3F: the authors performed a few experiments to exclude the involvement of the melanin receptor MelLec in the process. They stimulated macrophages with chemically synthesized 1,8-DHN and "observed that the levels of lactate secreted by macrophages stimulated with 1,8-DHN were significantly lower than those obtained after infection with the $\Delta ku80$ strain (Figure 3F), suggesting that recognition of 1,8-DHN melanin and signalling via MelLec is, to a large extent, dispensable for the activation of glycolysis." However, this experiment is not convincing: it would be necessary to perform a dose-response curve. Moreover, the results would be more convincing in the presence of data showing MelLec triggering in the experimental conditions used by the authors.*

Author's reply: Following the reviewer's suggestion, we have performed a dose-response curve of macrophages stimulated with 1,8-DHN. In these conditions, the levels of lactate secreted by stimulated macrophages were significantly lower than those obtained after infection with *A. fumigatus* (Fig. 8 of this reply letter). Protein levels of the receptor were not assessed due to the lack of specific antibodies. However, since murine macrophages do not express MelLec (Stappers et al, Nature 2018), and because

similar results were obtained with both human and murine macrophages regarding their metabolic reprogramming after infection, we are confident that our claim that signaling via MelLec is largely dispensable for glycolysis is correct.

Figure 8. Lactate secretion by human macrophages infected with the $\Delta ku80$ strain of *A. fumigatus* (MOI 1:10) or stimulated with different concentrations of 1,8-DHN (50, 75, 100 and 200 $\mu\text{g/mL}$) for 24 hours.

Reviewer's comment: Figure 4 and HIF-1 α : Fig 4E shows immunofluorescence in very few cells (from 1 to 3 per condition) and the results must to be quantified and expressed as HIF staining in nuclei/total HIF staining. Also, Fig S4E doesn't show significant inhibition of HIF-1 α accumulation in lysates. Considering these data, the involvement of HIF-1 α is questionable. Fig 4F is potentially very strong but proper controls are missing: it is important to show that HIF c macrophages recognize $\Delta ku80$ (phagocytosis). Moreover, is the expression of MelLec equivalent?

Author's reply: Following the reviewer's suggestion, we have now quantified and expressed as nuclear/total HIF-1 α using the image analysis software ImageJ (new Supplementary Fig. 4e of the revised manuscript). This information has also been added to the Methods section, under *Live cell imaging*. We agree that the western blot for HIF1 α may not have the best quality, but we respectfully disagree with the reviewer regarding that there are no differences. Although the differences are not outstanding (likely because we have analyzed total HIF1 α and not nuclear vs cytosolic HIF1 α), the pixel density quantification demonstrated different levels of total HIF-1 α , particularly in donor A. In addition, and as can be observed in Figure 9 of this reply letter, both wild type and HIF c macrophages were able to efficiently recognize and ingest the $\Delta ku80$ strain of *A. fumigatus*. Regarding MelLec expression, and as stated above (Stappers et al, Nature 2018), murine macrophages do not express this receptor. Therefore, our results clearly

demonstrate the involvement of HIF-1 α in the metabolic reprogramming of macrophages after *A. fumigatus* infection.

Figure 9. Bone marrow-derived macrophages (BMDMs) from wild-type and HIF c mice infected with FITC-labelled resting conidia of the $\Delta ku80$ strain of *A. fumigatus* (green) at a MOI of 1:10 for 4 and 8 hours.

Reviewer's comment: Figure 5: this figure is crucial for the manuscript but – again – lacks proper controls.

What about EGTA, TG, W7 in cells stimulated by $\Delta ku80$? Does calcium manipulation affect recognition of pathogens? And phagocytosis? Probably yes, so how to interpret the results shown in this figure? EGTA 500 μM to buffer extracellular calcium in DMEM is inadequate. Unless calcium-free medium is used, the experiments should be performed using 5mM EGTA. Was the pH controlled? Which statistical analysis was used in each figure 5 and why? (especially 5D, F, G).

Author's reply: As stated above, we agree that proper controls are crucial for improving the quality of the manuscript and we have now modified Fig. 5 to include the effects of calcium manipulation in cells stimulated by $\Delta ku80$. In addition, no significant effects of the treatments on phagocytosis were observed in our experimental conditions (Figure 10 of this reply letter). In line with the reviewer's suggestion, we attempted to perform the analyses using 5 mM of EGTA. In this case however, the viability of the cells was compromised due to the long period of infection (24h). As an alternative, we have used calcium-free medium alone and found that lactate secretion was significantly reduced when macrophages were infected with $\Delta pksP$ conidia recapitulating the previous results obtained with EGTA and excluding a possible role for extracellular calcium in the immunometabolic reprogramming of macrophages. Furthermore, we apologize for the possible misleading information regarding the experimental conditions used in Fig. 5. All experiments using thapsigargin and W7 were performed in calcium-free medium. We have now clarified this information further in the *Methods* section under *Cell stimulations*. In what concerns the mentioned figures, we have used a two-tailed t-test analysis, since our main goal was to evaluate a reversion of the pksP phenotype. Finally, we apologize to the reviewer for not having provided a section in the *Methods* describing the statistical tests used. Depending on the comparisons of interest, either two-tailed t-tests, one-way ANOVA or two-way ANOVA tests were used. We have now added this information to the *Methods* section under *Statistical analyses*.

Figure 10. Primary human macrophages were infected with FITC-labelled resting conidia of the $\Delta ku80$ strain of *A. fumigatus* at a MOI of 1:5 for 1 hour. When indicated, cells were pre-treated for 30 minutes with 2 μM of TG or treated with 25 μM of W7 added 10 minutes post-infection.

Reviewer's comment: *In Fig 5F the immunofluorescence doesn't make any point and the arrows in the first panel do not show mTOR accumulation. Concerning the image analysis, it is stated: "Data on quantification of mTOR+ phagosomes was determined by analyzing at least 200 phagosomes." But how was the analysis performed?*

Author's reply: The finding that mTOR localizes to the phagosome of macrophages infected with the $\Delta ku80$, but not $\Delta pksP$, strain, and that this defect is rescued after inhibition of calcium signaling highlights a molecular link between both processes leading to the activation of glycolysis. We are certain therefore that these results are highly relevant. To further clarify our point, we have highlighted the phagosomes with mTOR accumulation (as revealed by the rim of fluorescence) in the different experimental conditions using white arrows. In addition, the procedure for the quantification of mTOR+ phagosomes is described in the Methods section, under *Live cell imaging*. Briefly, phagosomes surrounded by a rim of fluorescence of the indicated protein marker were scored as positive, according to previous publications (Kymizi et al Nat Microbiol, 2018; Andrianaki et al Nat Commun 2018; Akoumianaki et al Cell Host Microbe 2016).

REVIEWERS' COMMENTS:

Reviewer #1 (Remarks to the Author):

I am happy that my previous comments have been appropriately addressed

Reviewer #2 (Remarks to the Author):

The authors have made extensive and well judged changes to the manuscript involving appropriate experimentation where needed. The manuscript is now suitable for publication.

Reviewer #3 (Remarks to the Author):

I am satisfied with the author revisions and responses to the specific points raised.

Reviewer #4 (Remarks to the Author):

The authors have addressed most of my questions and improved the manuscript.

Point-by-point reply letter

Reviewer #1

Reviewer's comments: *I am happy that my previous comments have been appropriately addressed.*

Reviewer #2

Reviewer's comments: *The authors have made extensive and well-judged changes to the manuscript involving appropriate experimentation where needed. The manuscript is now suitable for publication.*

Reviewer #3

Reviewer's comments: *I am satisfied with the author revisions and responses to the specific points raised.*

Reviewer #4

Reviewer's comments: *The authors have addressed most of my questions and improved the manuscript.*

Author's reply: We thank the reviewers for their comments during the revision that have helped us to improve the manuscript.